# Constraining global aerosol emissions using POLDER/PARASOL satellite remote sensing observations

Cheng Chen[1, 5], Oleg Dubovik[1], Daven K. Henze[2], Mian Chin[3], Tatyana Lapyonok[1], Gregory L. Schuster[4], Fabrice Ducos[1], David Fuertes[5], Pavel Litvinov[5], Lei Li[1, 6], Anton Lopatin[5], Qiaoyun Hu[1], Benjamin Torres[1]

[1]Laboratoire d'Optique Atmosphérique (LOA), UMR8518 CNRS, Université de Lille, Villeneuve D'ASCQ, 59655, France

[2]Department of Mechanical Engineering, University of Colorado, Boulder, Colorado, 80309, USA

[3]NASA Goddard Space Flight Center, Greenbelt, Maryland, 20771, USA

[4]NASA Langley Research Center, Hampton, Virginia, 23681, USA

[5]GRASP-SAS, Remote Sensing Developments, Université de Lille, Villeneuve D'ASCQ, 59655, France

[6]State Key Laboratory of Severe Weather (LASW) and Institute of Atmospheric Composition, Chinese Academy of Meteorological Sciences, CMA, Beijing, 100081, China

*Correspondence to*: Cheng Chen (cheng.chen@univ-lille.fr) and Oleg Dubovik (oleg.dubovik@univ-lille.fr)

**Abstract.** We invert global black carbon (BC), organic carbon (OC) and desert dust (DD) aerosol emissions from POLDER/PARASOL spectral aerosol optical depth (AOD) and aerosol absorption optical depth (AAOD) using the GEOS-Chem inverse modelling framework. Our inverse modeling framework uses standard *a priori* emissions to provide *a posteriori* emissions that are constrained by POLDER/PARASOL AODs and AAODs. The following global emission values were retrieved for the three aerosol components: 18.4 Tg/yr for BC, 109.9 Tg/yr for OC, and 731.6 Tg/yr for DD for the year 2010. These values show a difference of +166.7%, +184.0%, and -42.4% with respect to the *a priori* values of emission inventories used in "standard" GEOS-Chem runs. The model simulations using *a posteriori* emissions (i.e. retrieved emissions) provide values of 0.119 for global mean AOD and 0.0071 for AAOD at 550 nm, which are +13.3% and +82.1% higher than the AOD and AAOD obtained using the *a priori* values of emissions. Additionally, the *a posteriori* model simulation of AOD, AAOD, single scattering albedo, Ångström exponent, and absorption Ångström exponent show better agreement with independent AERONET, MODIS, and OMI measurements than the *a priori* simulation. Thus, this study suggests that using satellite-constrained global aerosol emissions in aerosol transport models can improve the accuracy of simulated global aerosol properties.

## 1 Introduction

Atmospheric aerosol emission inventories are often used to drive chemical transport model (CTM) simulations of aerosol distributions on regional and global scales (Boucher, 2015; Brasseur and Jacob, 2017; Granier et al., 2011). Satellite-retrieved columnar aerosol optical depth (AOD) is directly related to light extinction due to the presence of aerosols; hence, satellite-retrieved columnar AOD is widely used to evaluate the spatial and temporal variability of aerosols simulated from CTMs (e.g., Chin et al., 2002; Ginoux et al., 2006; Kinne et al., 2006, 2003; Liu et al., 2012; Ocko and Ginoux, 2017; Pozzer et

al., 2015; Schulz et al., 2006; Tegen et al., 2019). A general agreement has been shown for columnar AOD between model simulations and satellite observations in the Aerosol Comparisons between Observations and Models (AeroCom) "Experiment A" multi-model assessments (Kinne et al., 2006). However, this study also revealed large model diversity of species-specific AOD and aerosol absorption optical depth (AAOD), which encourages research to harmonize and improve the emissions of individual

aerosol species and aerosol precursors, representation of aerosol absorption and other elements (Kinne et al., 2006; Samset et al., 2018). Accurate knowledge of spatial and temporal distribution of species-specific aerosol emissions is also useful for numerical weather prediction (NWP) (Benedetti et al., 2018; Xian et al., 2019).

      There have been several efforts to improve aerosol emission inventories by using satellite observations and inverse modelling (Dubovik et al., 2008; Huneeus et al., 2012, 2013; Wang et al., 2012; Xu et al., 2013; Zhang et al., 2015; Escribano et

al., 2016, 2017; Zhang et al., 2005). However, most of them are regional studies that focus on a single aerosol or gas species, or use pre-defined regions to reduce the size of state vector (Huneeus et al., 2012; Zhang et al., 2015, 2005). There are only a few studies that use detailed satellite information to simultaneously retrieve emissions of multiple aerosol components at the native spatial resolution of a forward CTM model. For example, we have previously developed an inverse modelling framework for retrieving aerosol emissions of black carbon (BC), organic carbon (OC), and desert dust (DD) components within the GEOS-

Chem model (Chen et al., 2018). Specifically, the emissions of BC, OC and DD were simultaneously derived from satellite retrievals of spectral AOD and AAOD that were provided by the PARASOL (Polarization & Anisotropy of Reflectances for Atmospheric Sciences coupled with Observations from a Lidar) products. Chen et al. (2018) successfully used this method at the regional scale over all of Africa and the Arabian Peninsula.

      This work expands the inversion algorithm of Chen et al. (2018) to the global scale. We also refine an assumption that

defines the observation error covariance matrix for the recently released PARASOL Level 3 AOD and AAOD aerosol products generated by the GRASP (General Retrieval of Atmosphere and Surface Properties) algorithm. The method is applied to derive global BC, OC and DD aerosol emissions for the year 2010, using the updated Hemispheric Transport of Atmospheric Pollution (HTAP) Phase 2 emission data (HTAP, 2010; Janssens-Maenhout et al., 2015) for the initial estimate of anthropogenic emissions. Then the satellite-derived global *a posteriori* aerosol emissions are intensively evaluated in comparison to the *a priori* emission

inventories and other independent measurements.

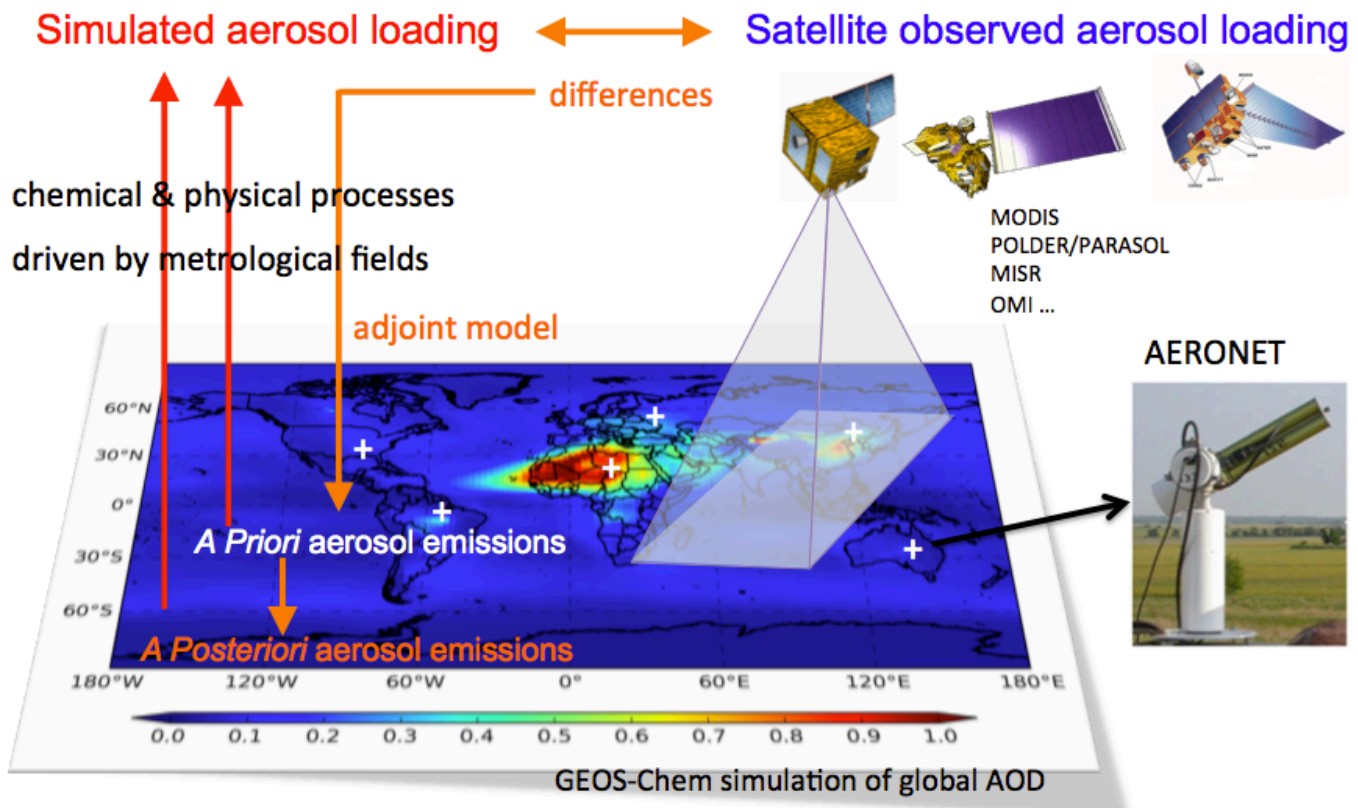

**Figure 1: General concept of satellite remote sensing of global aerosol emissions**

## 2 Methods and data

Figure 1 demonstrates the general concept of satellite remote sensing of the global distribution and strength of aerosol emissions. A priori emissions are used with GEOS-Chem model to create a simulated aerosol loading, which are then compared to observed spectral AOD and AAOD from PARASOL. The inverse modelling based on adjoint GEOS-Chem iteratively optimizes a priori emissions to minimize the differences between observed and modeled AOD and AAOD. A posteriori emissions are equivalent to retrieved or optimized emissions, which are then input to GEOS-Chem model simulation, and the

results are verified with independent AERONET, MODIS and OMI aerosol products.

### 2.1 POLDER/PARASOL aerosol dataset generated by the GRASP algorithm

       The GRASP algorithm implements statistically optimized fitting of diverse observations using the multi-term LSM (Least Square Method) method (Dubovik, 2004). The basic concept of this approach was introduced and implemented in the AERONET algorithm developed for aerosol characterization from ground-based radiometric observations (Dubovik and King, 2000; Dubovik et al., 2000; King and Dubovik, 2013). Dubovik et al. (2011, 2014) have adapted and extended this concept in the

GRASP algorithm, which is designed to retrieve aerosol and surface properties from satellite and other observations. As a new

inversion development, a multi-pixel retrieval concept was implemented in the GRASP (Dubovik et al., 2011). Using this concept, the satellite retrieval is implemented as a statistically optimal simultaneous fitting of observations over a large number of pixels. This approach allows for improved accuracy of the retrieval by applying known a priori constraints on temporal or/and spatial variability of the derived parameters. In addition, GRASP is a highly versatile algorithm that has been applied for a large variety of different types of satellite, ground-based, and airborne remote sensing measurements by photometers, lidars, satellite sensors, nephelometers, sky-cameras, etc. (Benavent-Oltra et al., 2017; Espinosa et al., 2017; Hu et al., 2019; Li et al., 2019; Lopatin et al., 2013; Román et al., 2017, 2018; Torres et al., 2017; Tsekeri et al., 2017).

The space borne multi-directional, multi-spectral polarized POLDER-3 (Polarization and Directionality of the Earth's Reflectance) imager on board PARASOL can measure the global angular distribution of intensity and polarization of solar radiation reflected to space by the earth-atmosphere system (Deschamps et al., 1994; Deuzé et al., 1999, 2001; Tanré et al., 2011). Throughout this article, we use "PARASOL" to denote POLDER/PARASOL observations. The GRASP algorithm inverts PARASOL comprehensive measurements to derive aerosol properties (e.g. extinction, absorption, size and composition) and surface BRDF (Bidirectional Reflectance Distribution Function) and BPDF (Bidirectional Polarization Distribution Function) properties. The development of the GRASP algorithm is described in Dubovik et al. (2011) and Dubovik et al. (2014), and a description of some PARASOL/GRASP aerosol products can be found in Chen et al. (2018), Kokhanovsky et al. (2015), Popp et al. (2016) and Sayer et al. (2018). The accuracy of the GRASP algorithm configured for AERONET measurements has been evaluated with a laboratory experiment by Schuster et al. (2019).

We use PARASOL/GRASP Level 3 AOD and AAOD at 6 wavelengths (443, 490, 565, 670, 865 and 1020 nm) in this study. The Level 3 PARASOL/GRASP aerosol products are rescaled in 1°x1° degree spatial resolution, and have been archived at the AERIS/ICARE Data and Services Center (http://www.icare.univ-lille1.fr). Information about PARASOL/GRASP Level 3 products is also available from the GRASP-OPEN site (https://www.grasp-open.com). In order to co-locate with model 2° (latitude) x 2.5° (longitude) spatial resolution, the PARASOL/GRASP aerosol data are aggregated into 2°x2.5° model grid boxes. Any grid box with less than two PARASOL/GRASP retrievals is omitted. The global distribution of PARASOL/GRASP 2°x2.5° spectral AOD and AAOD at 443, 490, 565, 670, 865 and 1020 nm used in inversion as presented in the supplement illustrations in Figure S1.

## 2.2 GEOS-Chem inverse modelling framework

The GEOS-Chem chemical transport model simulates the spatial and temporal mass distribution of each aerosol species by modelling transport processes (e.g. advection, convection, diffusion, deposition, etc.) and source injection (Bey et al., 2001; Brasseur and Jacob, 2017; Jacob, 1999). Once the global distribution of aerosol mass is known, it is generally converted to distributions of aerosol extinction (AOD) and absorption (AAOD) by modelling the aerosol microphysical and optical properties

(Martin et al., 2003). For this study, we simulate five major aerosol components, including sulfate (SU), BC, OC, DD (7 bins, with effective radii of $r_e = 0.14, 0.24, 0.45, 0.80, 1.40, 2.40,$ and $4.50\ \mu m$), and sea salt ($r_e = 0.80\ \mu m$ for the accumulation mode and $r_e = 5.73\ \mu m$ for the coarse mode). The microphysics of each aerosol species used in the simulation are given in Chen et al. (2018). Previously, we adopted two prevalent assumptions of BC refractive indices (Case 1:

$m = 1.75 - 0.45i$; Case 2: $m = 1.95 - 0.79i$). As suggested by the sensitivity test by Chen et al. (2018) using these two different assumptions leads to an additional factor ~2.0 differences for total retrieved BC emissions, because of the differences of the mass absorption efficiency ($\beta_a$; Case 1: $\beta_a = 4.5\ m^2 g^{-1}$; Case 2: $\beta_a = 6.3\ m^2 g^{-1}$ at 565 nm). However, the recommended value for uncoated BC at 550 nm is $7.5 \pm 1.2\ m^2 g^{-1}$ (Bond and Bergstrom, 2006). Hence, in this study, we retrieve BC emissions using $\beta_a$ from the Case 2 assumption. The details of microphysical properties of BC, OC, DD, SU and SS used in the

inversion are described in Chen et al. (2018). The assumption of external mixing of spherical particles is adopted in our inversion, as it is commonly done in most of CTMs. It should be noted, however, that the particle morphologies and mixing state could have strong affects on scattering and absorption properties, thus affecting mass to optical conversion (Liu and Mishchenko, 2018). For example, the "lensing effect" of less absorbing components coated on BC could amplify total aerosol absorption (Lesins et al., 2002). The absorption enhancement due to coating is estimated ~1.5 (Bond and Bergstrom, 2006). Recent study by

Curci et al. (2019) implemented partial internal mixing for regional simulation, and found it could improve simulation of total absorption while the spectral dependence can not be well reproduced. Therefore, in this approach, as well as, generally in CTMs there is some intrinsic ambiguity in assumptions influencing efficiency of scattering and absorption of aerosol particles. This ambiguity is certainly among of major factors affecting accuracy of derived emissions in the current approach.

      The inverse modelling framework used here was originally presented by Chen et al. (2018), wherein BC, OC, and DD

aerosol emissions were estimated simultaneously using spectral AOD and AAOD observations. The framework uses an adjoint of the GEOS-Chem model that was developed by Henze et al. (2007, 2009) and Wang et al. (2012). Here, SU and SS emissions are kept fixed (similar to Chen et al., 2018); in future studies, we plan to retrieve SU and SS emissions together with other emissions by using additional spatial and temporal (smoothness) constraints (Dubovik et al., 2008).

      Our inverse modelling method iteratively seeks adjustments of aerosol emissions that can minimize the cost function

$J(\boldsymbol{S})$,

$$J(\boldsymbol{S}) = \frac{1}{2}(\mathbf{H}(\boldsymbol{S}) - \boldsymbol{f}_{obs})^T \mathbf{C}_{obs}^{-1}(\mathbf{H}(\boldsymbol{S}) - \boldsymbol{f}_{obs}) + \frac{1}{2}\gamma_r(\boldsymbol{S} - \boldsymbol{S}_a)^T \mathbf{C}_a^{-1}(\boldsymbol{S} - \boldsymbol{S}_a). \tag{1}$$

In Eq (1), the forward model $\mathbf{H}$ is a CTM (e.g. GEOS-Chem). $\boldsymbol{f}_{obs}$ is the vector of observed parameters used for inversion, $\mathbf{C}_{obs}$ is the error covariance matrix of $\boldsymbol{f}_{obs}$. The vector $\boldsymbol{S}$ describes the 4D distribution of emissions, and $\boldsymbol{S}_a$ is the *a priori* estimates of emissions. $\mathbf{C}_a$ is the error covariance matrix of $\boldsymbol{S}_a$. $\gamma_r$ is a regularization parameter.

The inversion is initialized using *a priori* model emissions plus a spatially uniform value of $10^{-4}$ for DD, $10^{-6}$ for BC and $5\times10^{-6}$ for OC over land grid boxes where $\boldsymbol{S}_a = 0$; this allows the detection of new sources and performs satisfactorily even at locations where the a priori knowledge of aerosol emission is poor (Chen et al., 2018). In this study, the a priori DD emission is based upon the mineral dust entrainment and deposition (DEAD) scheme (Zender et al., 2003) and GOCART dust source function (Ginoux et al., 2001) that was implemented in GEOS-Chem model by Fairlie et al. (2007). We adopted the improved fine mode dust contribution scheme by Zhang et al. (2013) for both inversion and simulation. We used anthropogenic emissions from the Hemispheric Transport of Atmospheric Pollution (HTAP) Phase 2, with biomass burning emissions from version 4s of the Global Fire Emissions Database (GFED; Randerson et al., 2012; van der Werf et al., 2017). The GEOS-Chem sulfate module was developed by Park et al. (2004) and carbonaceous aerosol simulations by (Park et al., 2003), both of which are based upon the GOCART model scheme (Chin et al., 2002). The sea salt simulation is based upon Jaeglé et al. (2011).

The inversion system derives daily total BC, OC and DD aerosol emissions for each grid box. The daily ratio between biomass burning and anthropogenic contribution for BC and OC and the proportion of DD 7 bins for each grid box is kept as a priori GEOS-Chem assumption. Distinguishing anthropogenic contribution from total emission is crucial for climate effects evaluation. Here, we propose a simple method to estimate daily anthropogenic BC ($E_{BC\_INV\_AN}$) and OC ($E_{OC\_INV\_AN}$) emission from our retrieved total emission ($E_{BC\_INV}$) by using daily proportion of anthropogenic emission over each grid box from a priori emission database:

$$E_{BC\_INV\_AN}(x,y,t) = \frac{E_{BC\_AN}(x,y,t)}{E_{BC\_AN}(x,y,t) + E_{BC\_BB}(x,y,t)} * E_{BC\_INV}(x,y,t) \tag{2}$$

$$E_{OC\_INV\_AN}(x,y,t) = \frac{E_{OC\_AN}(x,y,t)}{E_{OC\_AN}(x,y,t) + E_{OC\_BB}(x,y,t)} * E_{OC\_INV}(x,y,t) \tag{3}$$

where $E_{BC\_INV}(x,y,t)$ and $E_{OC\_INV}(x,y,t)$ represent retrieved total BC and OC emission. $E_{BC\_INV\_AN}(x,y,t)$ and $E_{OC\_INV\_AN}(x,y,t)$ are derived anthropogenic BC and OC emissions from retrieved total BC and OC emission database. $E_{BC\_AN}(x,y,t)$ and $E_{OC\_AN}(x,y,t)$ represent anthropogenic BC and OC emission from HTAP v2 database, $E_{BC\_BB}(x,y,t)$ and $E_{OC\_BB}(x,y,t)$ are BC and OC emitted from biomass burning adapted from GFED v4s database. $x, y$ and $t$ indicate indexes of longitude, latitude and time.

**2.3 Error covariance matrix**

The error covariance matrix $\mathbf{C}_{obs}$ of observations $\boldsymbol{f}_{obs}$ needs to be prescribed in our inversion system. In our previous study, we tested and adopted a spectral weighting scheme to fit the absolute value of AOD and AAOD at different wavelengths (Chen et al., 2018). Specifically, PARASOL/GRASP AOD and AAOD were rigorously evaluated against the AERONET dataset. The current understanding of the accuracy of PARASOL/GRASP products is ~0.05 for absolute AOD. The AAOD accuracy is

unknown but expected to be positively correlated with the AOD value; therefore, the AAOD accuracy is assumed to be 0.05/AOD. In this work, we assume the covariance error matrices ($\mathbf{C}_{obs}$ and $\mathbf{C}_a$) are diagonal, and use these accuracy estimates for AOD and AAOD as the diagonal terms of the observation covariance matrix $\mathbf{C}_{obs}$.

The error covariance matrix $\mathbf{C}_a$ for the a priori emission dataset $\boldsymbol{S}_a$ is not accurately known. As a result, this matrix is often defined using rather simple strategies that are based upon uncertainty estimates found in the literature (e.g. Escribano et al., 2016, 2017; Huneeus et al., 2012). In our previous study, we adopted a very small regularization parameter $\gamma_r$ (e.g. 1.0e-4) to force the inversion to rely upon the observations (Chen et al., 2018). Additionally, our previous simulations using the a posteriori emissions have shown good agreement with independent measurements, so we decided to use the same strategy for our global inversions in this study.

## 3 Results

### 3.1 Emissions

We applied our method to retrieve BC, OC and DD daily emissions on a global basis using PARASOL spectral AODs and AAODs for the year 2010. The retrieved emissions dataset are publicly available at the LOA website (http://www-loa.univ-lille1.fr/article/a/emissions-aerosols-echelle-globale-restituees-par-modelisation-inverse-Chen-et-al). Table 1 summarizes the annual a priori and a posteriori emissions of five aerosol components (DD, BC, OC, SU and SS). The a posteriori DD, BC and OC emissions are retrieved from PARASOL spectral AOD and AAOD, while the a posteriori SS emission and the SU sources (from $SO_2$ oxidation) remain the same as the a priori simulation. We break the one-year global inversion into 12 months, and the maximum of iterations is set to 40 for each individual retrieval run.

The retrieved aerosol emissions are 18.4 Tg/yr for BC, 109.9 Tg/yr for OC and 731.6 Tg/yr for DD in year 2010. These new emissions indicate an increase of 166.7% for BC and 184.0% for OC, while a decrease of 42.4% for DD with respect to the a priori GEOS-Chem emission database. Table 2 compares the retrieved annual anthropogenic and biomass burning BC and OC emissions with a priori emission database in GEOS-Chem. The method used to separate anthropogenic from total emission is described in Section 2.2. The retrieved anthropogenic emissions are 14.8 Tg/yr for BC and 85.6 Tg/yr for OC, representing an increasing of 217.3% for BC and 357.8% for OC. Meanwhile, the retrieved biomass burning emissions of BC and OC are 3.6 Tg/yr and 24.3 Tg/yr, corresponding to an increase of 56.5% and 21.5% with respect to the a priori emission database. The comparison of spatial distribution of anthropogenic and biomass burning emission of BC and OC are presented in the supplement illustrations in the Figures S5 and S6.

**Table 1. Total source strengths (unit: Tg/yr) of five major aerosol components in year 2010.**

|  | DD [a] | BC | OC | SO$_2$ [b, c] | SS [c] |
|---|---|---|---|---|---|
| *A priori* [d] | 1269.4 | 6.9 | 38.7 | 87.9 | 3540.3 |
| *A posteriori* [e] | 731.6 | 18.4 | 109.9 | 87.9 | 3540.3 |

[a] Estimated based upon a priori GEOS-Chem model emission database

[b] Retrieved emission database that are used for a posteriori GEOS-Chem model simulation

5    [c] The SO$_2$ and SS emission are kept the same as a priori GEOS-Chem emission database in this study

[d] Particle radius ranging from 0.1 to 6.0 $\mu m$

[e] Unit: Tg S/yr

**Table 2. Anthropogenic (AN) and biomass burning (BB) emissions (unit: Tg/yr) of BC and OC in year 2010**

|  | BC | | OC | |
|---|---|---|---|---|
|  | AN | BB | AN | BB |
| *A Priori* | 4.6 | 2.3 | 18.7 | 20.0 |
| *A Posteriori* | 14.8 | 3.6 | 85.6 | 24.3 |

      Spatial distributions of the a priori and a posteriori annual emissions of BC, OC, and DD and their differences (a posteriori minus a priori emissions) are shown in Figure 2a, 2b, and 2c for the year 2010. The a posteriori BC emissions are generally greater than the a priori BC emissions throughout the globe; the a posteriori increases are particularly significant over certain regions, such as Southeast Asia and central and northwest China. However, there are also notable decreases of BC a

15   posteriori emissions (with respect to a priori emissions) observed in several grid boxes over South America and North China. The largest increase of OC emissions is in Southern Africa, where biomass burning is the predominant source. Consistent with BC, there is also a slight decrease of OC emissions over the high emission grid boxes in South America. In contrast, the a posteriori DD emissions are reduced throughout the global desert regions. As a reference, the seasonal cycle of the a posteriori BC, OC and DD emissions are presented in the supplement illustrations in the Figures S2-S4.

20      Notably, the a posteriori BC and OC emission distributions are more homogenous than the a priori emission inventories. This phenomenon is probably because the emissions are reported for 2° x 2.5° grid boxes, and this is too coarse to characterize cities with high anthropogenic activities. The spread of a posteriori BC and OC emission distributions leads to more grid boxes being influenced by absorbing aerosols over India and China, which can be supported by evaluation with AERONET retrieval of

AAOD (see Section 3.3.1). Briefly, there is a set of AERONET data of moderate to high aerosol absorption (AAOD > ~0.07 at 550 nm) where the a priori simulation was very close to zero, and the a posteriori simulation adjusted them (see Section 3.3.1).

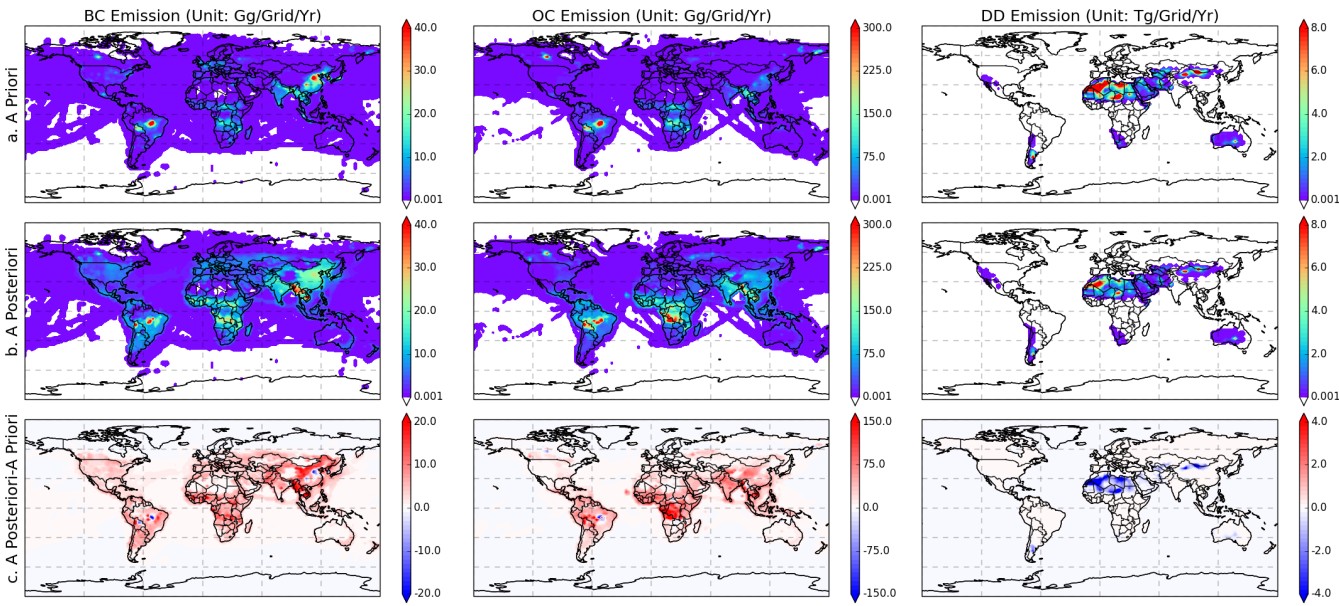

**Figure 2. Global distribution of emissions for 2010 for BC (left panel), OC (middle panel) and DD (right panel) based on (a) a priori and (b) a posteriori emission datasets; and the differences between a posteriori and a priori emissions (c).**

Comparisons of monthly global total BC, OC, and DD emissions between a priori and a posteriori emission inventories for 2010 are shown in Figure 3. Both of a priori and a posteriori BC and OC emission inventories show a maximum in August and September, while the second peak in March observed in the a priori database shift to April and May in the a posteriori database. However, a posteriori BC and OC emissions are higher than the a priori emissions throughout the year. The posteriori/priori ratio for monthly BC emissions is up to 4.4 in April and down to 1.1 in July. Meanwhile, the posteriori/priori ratio for monthly OC emissions is up to 4.6 in October and down to 1.4 in July. In contrast, the a posteriori DD emissions capture seasonal variations that are similar to the a priori DD emissions. The a posteriori DD emissions are reduced consistently throughout the year. The posteriori/priori monthly DD emissions ratio slightly varies between 0.51 (December) and 0.78 (June).

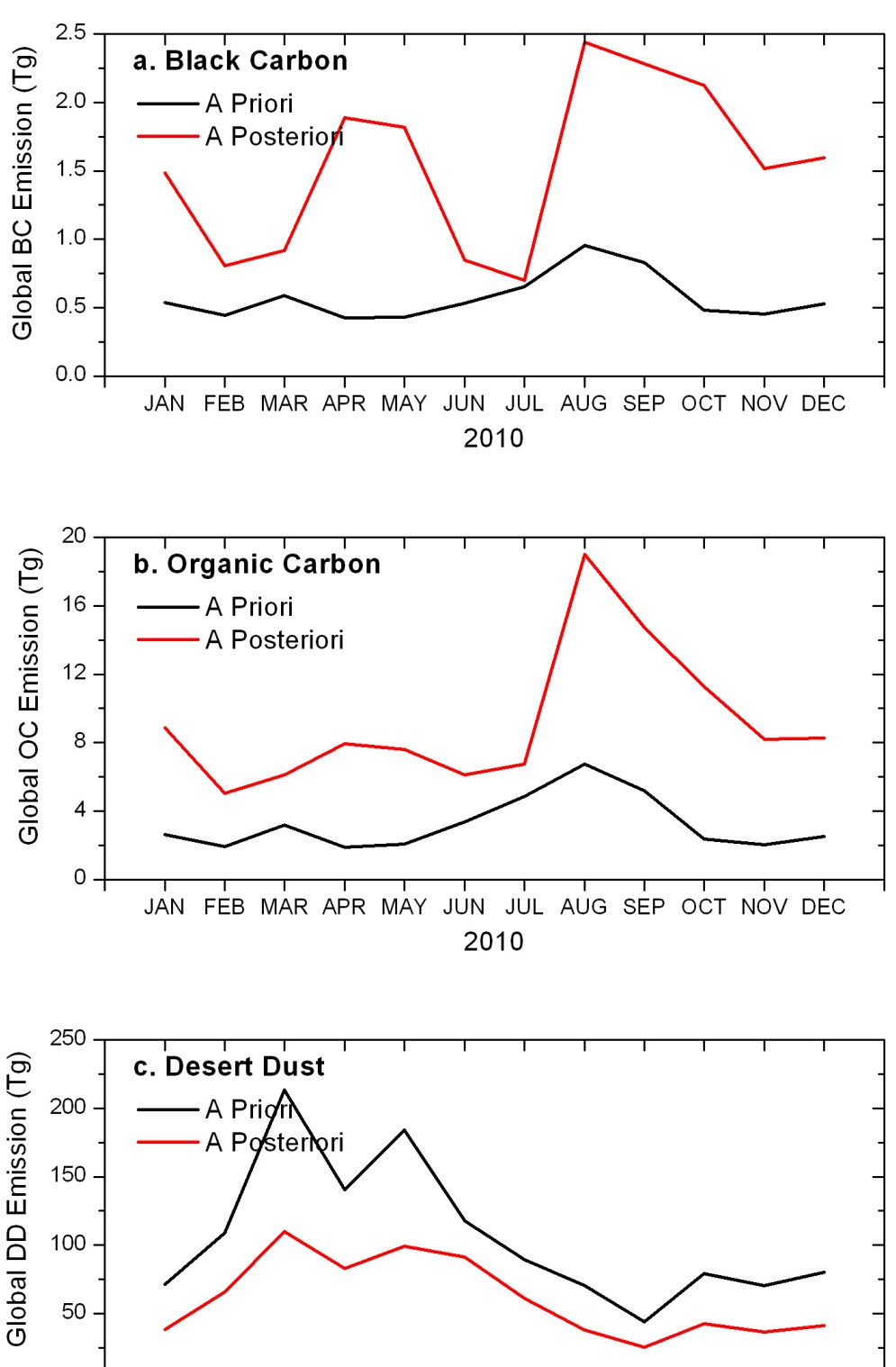

**Figure 3. Comparison of monthly emissions for a priori and a posteriori datasets: (a) BC emissions, (b) OC emissions, and (c) DD emissions.**

**3.2 Simulation of AOD and AAOD**

In a numerical modelling experiment for year 2010, we used the a priori and a posteriori emissions inventories as inputs to the GEOS-Chem model. Then we compared the simulated AODs and AAODs that were generated from the a priori emissions inventory to the AODs and AAODs that were likewise generated from the a posteriori emissions inventory. The results are shown in Figures 4 and 5 for AOD and AAOD at 550 nm.

Both simulations with the a priori (Figure 4a) and a posteriori (Figure 4b) emissions show that high values of AOD appear over the Sahara desert in North Africa and the Taklimakan and Gobi deserts in Asia, which are strong dust source regions. Figure 4 also indicates high values of AOD in East Asia, where anthropogenic aerosols (e.g. BC, OC, and SU) are the predominant components.

One of the major differences (Figure 4c) between a priori and a posteriori AOD is that the a posteriori AOD over desert regions is reduced, especially over the Sahara; meanwhile the a posteriori AOD increases over industry and biomass burning regions. The strongest increasing of a posteriori AOD occurs in southern Africa, where it is associated with biomass burning emissions.

Figure 5 shows the comparison of global distribution of a priori (Figure 5a) and a posteriori (Figure 5b) AAOD in 2010. Figure 5c clearly reveals that the a posteriori AAOD is higher than the a priori AAOD throughout the globe, except over the Sahara. The differences are high over southern Africa, India, Southeast Asia and central China, where they are associated with biomass burning and anthropogenic emissions.

The statistics of global annual mean AOD and AAOD at 550 nm for the five major aerosol components using a priori and a posteriori simulations are shown in Table 2. The a priori GEOS-Chem global mean AOD at 550 nm is 0.105, while the a posteriori simulation showed a slightly increased value of 0.119. The dust AOD decreases from 0.031 to 0.019 and its relative contribution to total AOD decreases from 29.9% to 16.1%, owing to the reduction of global DD emission from 1269.4 Tg/yr (a priori) to 731.6 Tg/yr (a posteriori). On the other hand, the a posteriori simulation indicates significant increase of carbonaceous AOD from a priori 0.014 (BC: 0.003; OC: 0.011) to a posteriori 0.040 (BC: 0.008; OC: 0.032). The resulting OC emissions increase from 38.7 Tg/yr to 109.9 Tg/yr, and OC becomes the largest contributor to the total AOD. With the a posteriori simulation, OC accounts for 26.8% of the total AOD, whereas the a priori simulation indicated DD as the largest contributor to total AOD (at 29.9%).

In general, the global mean AAOD at 550 nm significantly increases from 0.0039 (a priori) to 0.0071 (a posteriori), i.e., by a factor of 1.8. In particular, DD and BC are the two major components to the total aerosol absorption that collectively account for 90.1% (a priori) and 88.2% (a posteriori) of total AAOD. The a posteriori DD AAOD decreases by 42.8% relative to

the a priori simulation, from 0.0014 to 0.0008. The BC AAOD increases from 0.0020 (a priori) to 0.0054 (a posteriori), a factor of ~2.7, and the relative contribution to total AAOD increases from 52.7% to 76.9%.

In comparison to the AeroCom Phase II assessments of global AOD and AAOD (Myhre et al., 2013), our GEOS-Chem a posteriori global mean AOD is ~8% smaller. However, our a posteriori GEOS-Chem global mean AAOD (0.0071) is ~69% larger than the AeroCom multi-model mean of 0.0042±0.0019 (1 std. dev.).

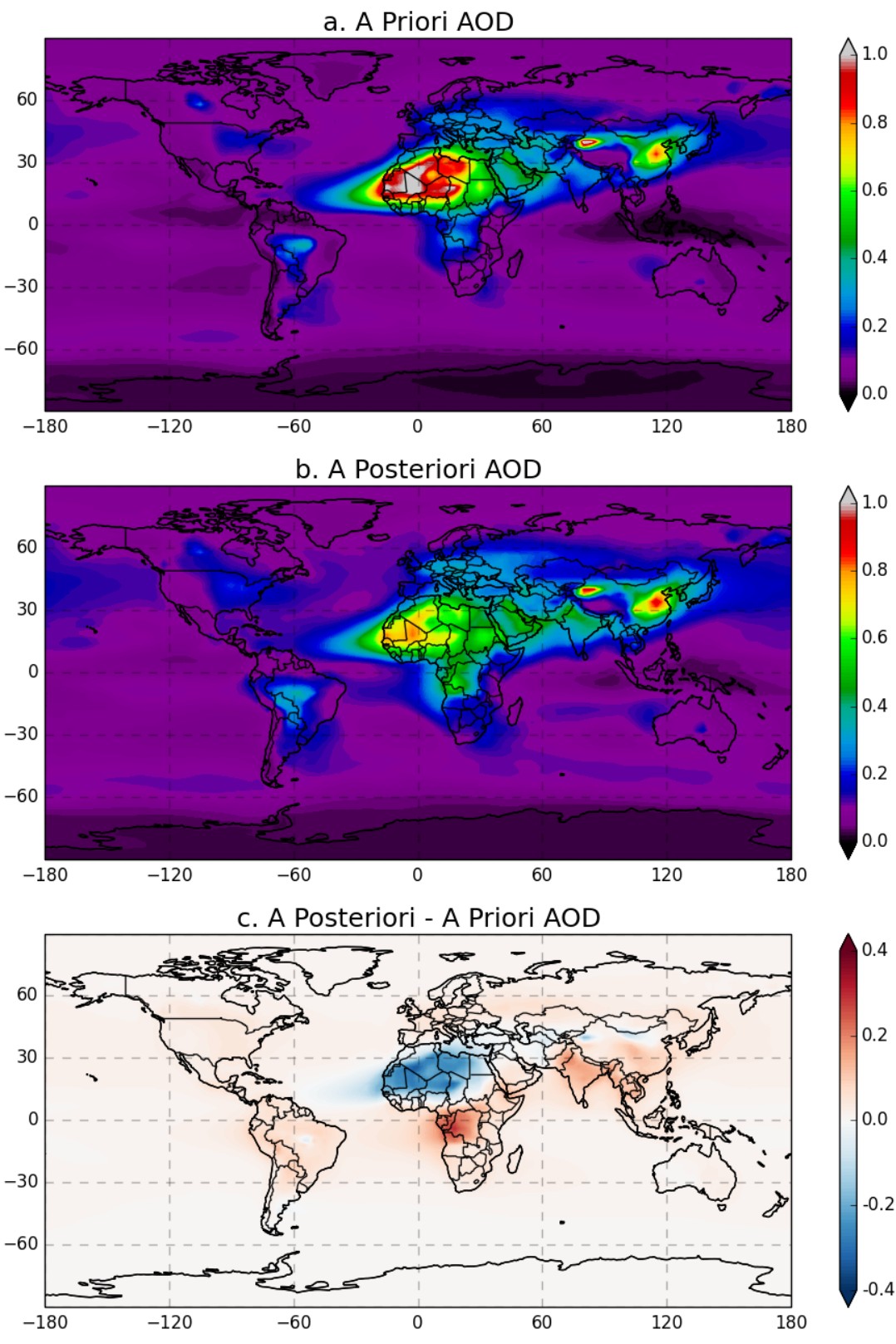

**Figure 4. Comparison of GEOS-Chem simulation of global aerosol optical depth in year 2010 at 550 nm based upon a priori and a posteriori emission datasets: (a) a priori AOD, (b) a posteriori AOD, and (c) a posteriori minus a priori AOD.**

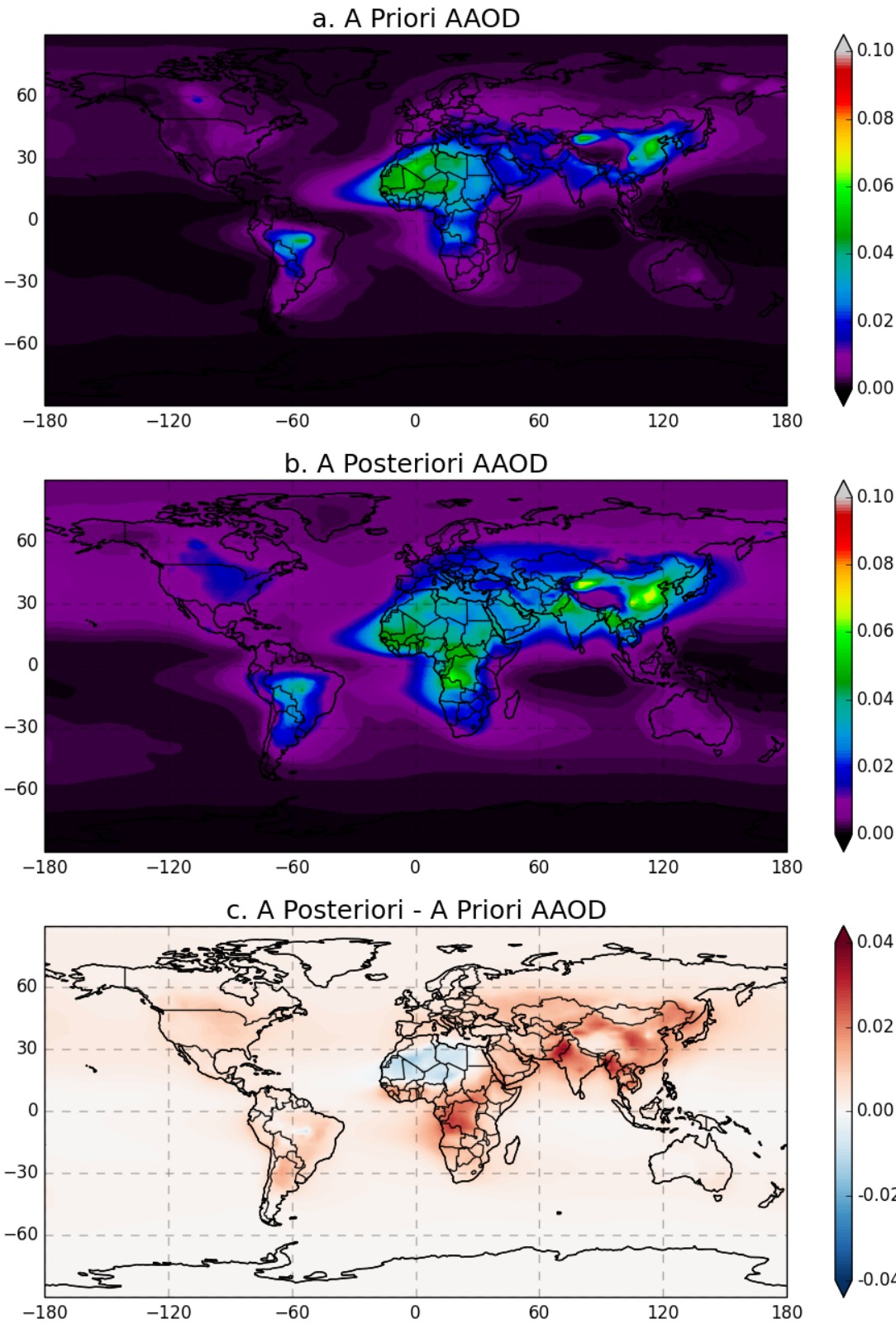

**Figure 5. Same as Figure 4, but for AAOD.**

**Table ~~23~~. Global mean AOD and AAOD at 550 nm of five major aerosol components. Numbers in parenthesis are relative contributions of different components to AOD and AAOD at 550 nm. Statistics are based upon a priori and a posteriori GEOS-Chem simulations for the year 2010.**

| | A priori | | A posteriori | |
|---|---|---|---|---|
| | AOD | AAODx10 | AOD | AAODx10 |
| DD | 0.031 (29.9%) | 0.014 (37.4%) | 0.019 (16.1%) | 0.008 (11.3%) |
| BC | 0.003 (2.7%) | 0.020 (52.7%) | 0.008 (6.5%) | 0.054 (76.9%) |
| OC | 0.011 (10.5%) | 0.002 (6.0%) | 0.032 (26.8%) | 0.007 (9.6%) |
| SU | 0.031 (29.2%) | 0.000 (0.8%) | 0.031 (26.0%) | 0.000 (0.5%) |
| SS | 0.029 (27.7%) | 0.001 (3.1%) | 0.029 (24.6%) | 0.001 (1.7%) |
| Total | 0.105 | 0.039 | 0.119 | 0.071 |

### 3.3 Evaluation with independent measurements

To achieve a more robust evaluation, the a priori and a posteriori aerosol properties simulated with the GEOS-Chem model are evaluated with other independent measurements that are not used in our emission inversion. The definition of the statistics used in the comparison, including correlation coefficient (R), root mean square error (RMSE), mean absolute error (MAE), normalized mean bias (NMB), and mean bias (MB), can be found below:

$$R = \frac{\sum(M_i - \bar{M})(O_i - \bar{O})}{\sqrt{\sum(M_i - \bar{M})^2(O_i - \bar{O})^2}} \qquad (4)$$

$$RMSE = \sqrt{\frac{\sum_{i=1}^{N}(M_i - O_i)^2}{N}} \qquad (5)$$

$$MAE = \frac{1}{N}\sum_{i=1}^{N}|(M_i - O_i)| \qquad (6)$$

$$NMB = \sum(M_i - O_i)/\sum O_i \times 100\% \qquad (7)$$

$$MB = \frac{1}{N}\sum_{i=1}^{N}(M_i - O_i) \qquad (8)$$

where $M$ represents the model results, $O$ represents the observations, $\bar{M}$ and $\bar{O}$ are the mean values for the simulations and observations, and $N$ is the number of data points.

### 3.3.1 Comparison with AERONET measurements

The Aerosol Robotic Network (AERONET; Holben et al., 1998) has provided comprehensive and accurate aerosol data from a worldwide ground-based sun-photometer network for more than two decades. The data products include measurements of multiple-wavelength AODs and Ångström exponents (AExp). Additionally, the AERONET products also include retrievals of AAOD, single scattering albedo (SSA), absorption Ångström exponent (AAExp), size distribution, and complex refractive index (Dubovik and King, 2000, Dubovik et al., 2000, 2002, 2006). The AERONET aerosol dataset have been commonly used for satellite product validation and model evaluation for a wide range of aerosol research topics. In this section, the AERONET version 2 daily level 2.0 (Smirnov et al., 2000) AOD, AAOD, SSA at 550 nm, AExp (440-870 nm), and AAExp (440-870 nm) products are used to evaluate the a priori and a posteriori GEOS-Chem model aerosol simulation. We convert spectral AOD and AAOD to SSA, AExp, and AAExp using the following equations:

$$SSA(\lambda) = 1.0 - \frac{AAOD(\lambda)}{AOD(\lambda)} \tag{9}$$

$$AExp = \ln\left(\frac{AOD(\lambda_1)}{AOD(\lambda_2)}\right) / \ln\left(\frac{\lambda_2}{\lambda_1}\right) \tag{10}$$

$$AAExp = \ln\left(\frac{AAOD(\lambda_1)}{AAOD(\lambda_2)}\right) / \ln\left(\frac{\lambda_2}{\lambda_1}\right) \tag{11}$$

We evaluate the GEOS-Chem model-simulated daily aerosol properties with the AERONET dataset. Globally, 282 AERONET sites that have available data for the year 2010 are all taken into account. We should note that AERONET aerosol datasets are averaged of all available data during the daytime (lunar photometer aerosol products will probably be available in the future; e.g. Barreto et al., 2016 and Berkoff et al., 2011), and that the GEOS-Chem daily aerosol properties are averaged based upon daytime and nighttime simulations. Additionally, the use of AERONET data comparing against corresponding 2° x 2.5° grid box model simulation could bring some sampling issues that cannot be ignored (Schutgens et al., 2016).

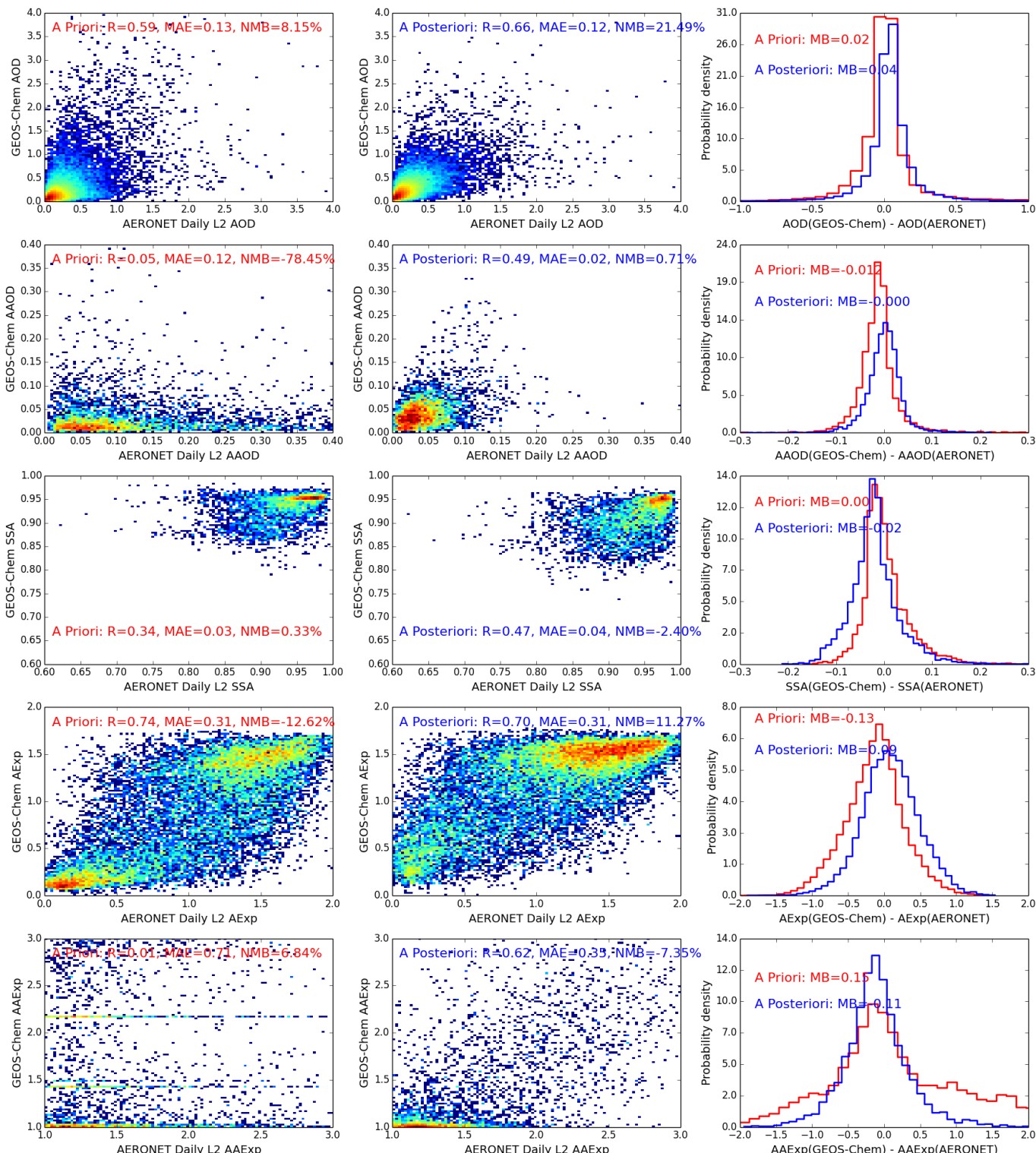

**Figure 6.** Comparison of a priori and a posteriori GEOS-Chem simulated AOD, AAOD, SSA at 550 nm, AExp (440-870 nm), and AAExp (440-870 nm) with AERONET Level 2 daily aerosol products. The data for density plot are all aggregated into 100 bins for both x- and y- directions spanning from minimum to maximum value in the axis. The correlation coefficient (R), mean absolute error (MAE), normalized mean bias (NMB), and mean bias (MB) are also provided on the panels.

Figure 6 shows the comparison of a priori (red) and a posteriori (blue) GEOS-Chem model simulated AOD, AAOD, SSA, AExp and AAExp with the AERONET dataset. The a posteriori GEOS-Chem simulation has much better agreement with AERONET data for AAOD. The correlation coefficient (R) of a priori GEOS-Chem AOD with AERONET equals to 0.59, mean absolute error (MAE) 0.13, and normalized mean bias (NMB) 8.15%. The a posteriori simulation improves R to 0.66 and MAE to 0.12. The NMB from a posteriori simulation (21.49%) is higher than a priori simulation, which indicates the a posteriori GEOS-Chem AOD is generally ~20% higher than the AERONET dataset. This is possibly due to the use of AERONET point measurements to evaluate model simulation at 2° x 2.5° grid box. The statistics for simulated AAOD shows significant improvement from the a priori (R=0.05, MAE=0.12, NMB=-78.45%, MB=-0.012) to the a posteriori (R=0.49, MAE=0.02, NMB=0.71%, MB=0.000) simulation. The a priori GEOS-Chem simulated AAOD is lower than AERONET data (NMB=-78.45%, MB=-0.012), while the a posteriori AAOD is much closer to AERONET (NMB=-0.71%, MB=0.000).

The inversion framework derives aerosol emissions by fitting spectral AOD and AAOD from PARASOL. The SSA, AExp and AAExp are the derived products from spectral AOD and AAOD (Eq. 9-11). The GEOS-Chem model SSA correlations with AERONET improve slightly from R=0.34 for the a priori to R=0.47 for the a posteriori simulation. The a priori and a posteriori simulations of AExp show similar performance when evaluating with AERONET data. However, the a posteriori R=0.70 for AExp is slightly worse than the a priori (R=0.74), which is likely related with ~20% high bias of the a posteriori AOD. Consistent with AAOD, the GEOS-Chem AAExp shows improvements from the a priori simulation (R=0.01, MAE=0.71, NMB=6.84%, MB=0.15) to the a posteriori simulation (R=0.62, MAE=0.35, NMB=-7.35%, MB=-0.11), which indicates a better representation of the spectral dependence of aerosol absorption (Russell et al., 2010; Schuster et al., 2016).

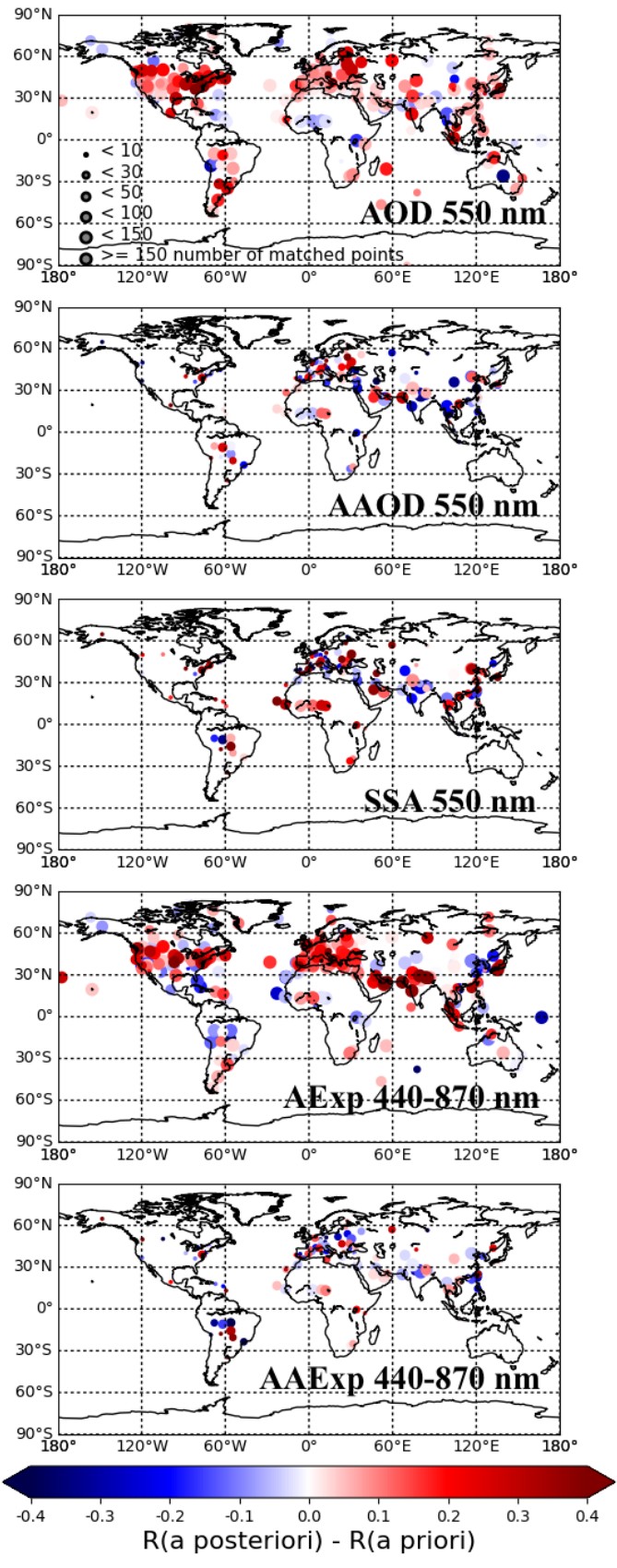

**Figure 7. The differences between a posteriori and a priori GEOS-Chem simulated AOD, AAOD, SSA, AExp and AAExp correlation coefficients (R) with AERONET daily aerosol products over all sites with collocated data in 2010**

To better understand the regional performance of a priori and a posteriori GEOS-Chem simulation, we conducted the comparison of daily aerosol products of AOD, AAOD, SSA, AExp and AAExp over all AERONET sites with collocated data in 2010. Figure 7 shows the differences of correlation coefficient between the a posteriori and a priori simulations. The red circles indicate sites where the a posteriori simulation has higher correlation with AERONET than the a priori simulation. Alternatively, the blue circles indicate sites where the a priori simulation shows better correlation than the a posteriori simulations. There are 202 sites out of a total of 282 sites that show improved correlation for AOD using the a posteriori emission data. Additionally, 176 out of 272 sites show improvement for AExp, 97 of 162 sites show improvement for SSA, 89 of 167 sites show improvement for AAOD, and 84 of 167 sits have a better correlation for AAExp. The a posteriori simulation loses correlation with AERONET for AAOD in central Eurasia and the western United States, which needs further investigation in future study.

Intensive wide fire events over central Russia during the Summer of 2010 have been reported in several studies (e.g. Chubarova et al., 2012; Gorchakova and Mokhov, 2012; Huijnen et al., 2012; Péré et al., 2014; R'honi et al., 2013). An increase of daily AOD is observed at the AERONET site Moscow_MSU_MO (55.707°N, 37.522°E) from late July to early August, with a maximum on 7 August 2010. The ground-based observation of this short-term fire event is suitable to evaluate the retrieved emissions used for daily simulation results. The comparison of the time series of the daily AOD and AAOD at 550 nm has been conducted for site Moscow_MSU_MO, and the results are shown in Figure 8. The geo-locations of Moscow_MSU_MO site are apparent in Figure 9 as a red star. The a priori GEOS-Chem simulation underestimates aerosol extinction and absorption during the fire event, which may be associated with underestimation of biomass burning emissions. The a posteriori GEOS-Chem simulation has much better agreement with observations in terms of temporal variation and intensity using retrieved daily BC and OC emissions.

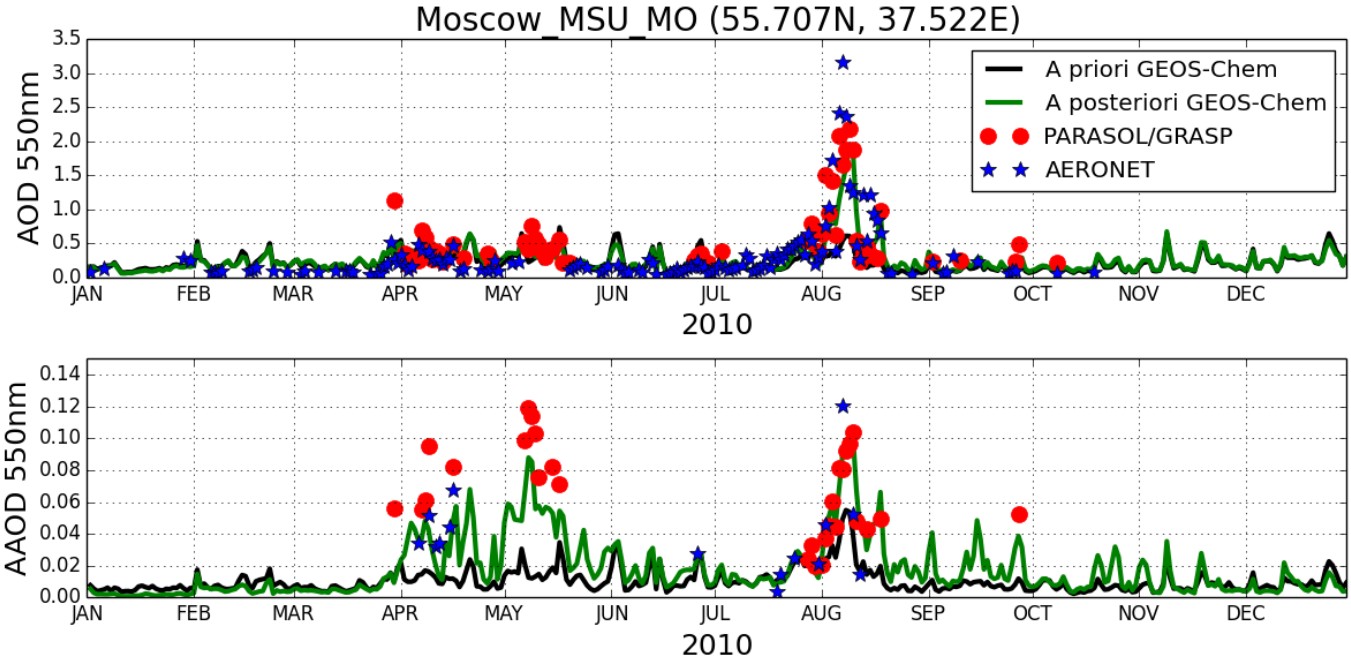

**Figure 8. Time serial plot of AOD and AAOD at 550nm from AERONET (blue star), PARASOL/GRASP (red circles), a priori (black line) and a posteriori (green line) GEOS-Chem simulations at Moscow_MSU_MO (55.707°N, 37.522°E) site**

### 3.3.2 Comparison with MODIS aerosol products

In this section, we evaluate agreement of a priori and a posteriori GEOS-Chem simulated AOD at 550 nm with Moderate Resolution Imaging Spectroradiometer (MODIS) Collection 6 (C6) Level 3 merged aerosol products (Sayer et al., 2014). Dark Target (DT) and Deep Blue (DB) are two well known retrieval algorithm developed for processing MODIS atmospheric aerosol products suite (Hsu et al., 2004, 2013; Kaufman et al., 1997; Levy et al., 2013; Remer et al., 2005; Sayer et al., 2013; Tanré et al., 1997). The MODIS/Aqua merged aerosol products combing the DB with DT land/ocean data provide more gap-filled retrievals, which is suitable for model evaluation. In order to collocate with model data, the 1 degree Level 3 products are then aggregated into model 2° x 2.5° grid box, and any grid box with less than 2 MODIS AOD is omitted.

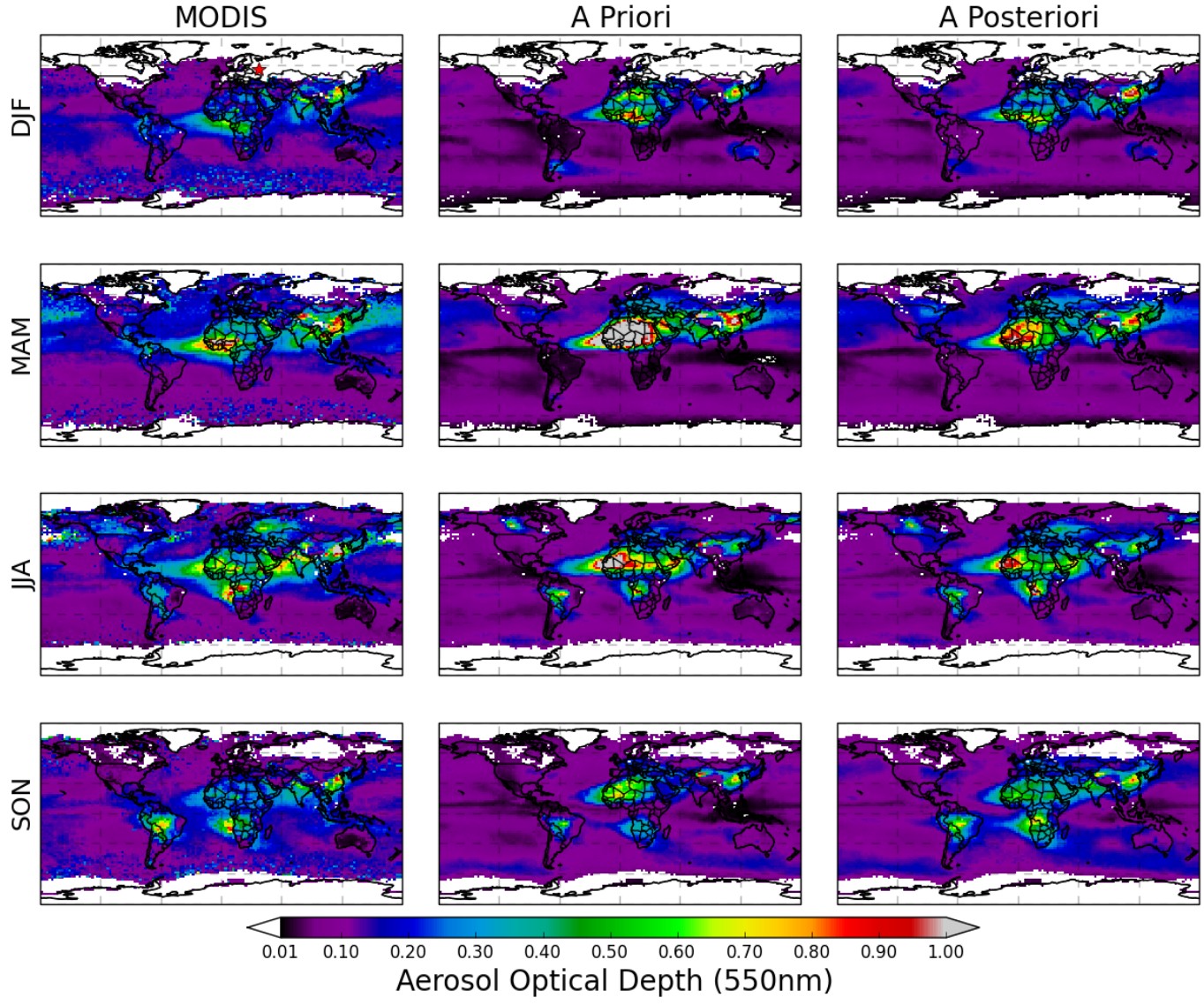

**Figure 9. Comparison of a priori and a posteriori GEOS-Chem simulated seasonal AOD at 550 nm with MODIS C6 Dark Target-Deep Blue merged products**

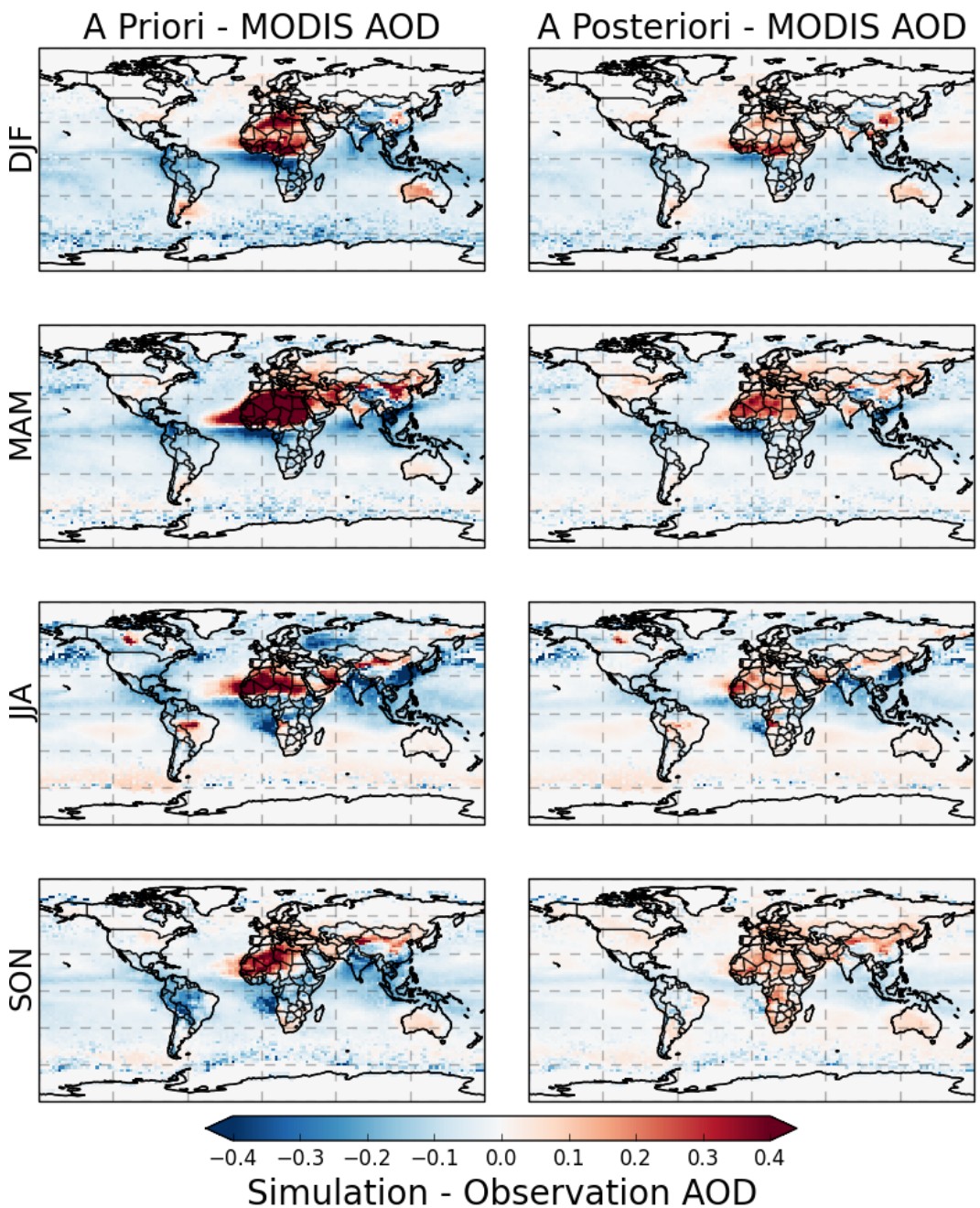

**Figure 10. Differences between a priori and a posteriori GEOS-Chem seasonal AOD with MODIS C6 merged products**

In order to assess the aerosol seasonal cycle and some peaks of aerosol loading, we focus on AOD seasonal pattern from

5    MODIS, a priori and a posteriori GEOS-Chem simulations (Figure 9) as well as their differences (Figure 10). The a priori

GEOS-Chem simulation strongly overestimates aerosol loading over Sahara throughout the year; the aerosol loading over Sahara

is reduced in the a posteriori GEOS-Chem simulation, while the simulated AOD is still slightly higher than the MODIS AOD.

Over India, the a priori simulation underestimates aerosol loading in December-January-February (DJF), June-July-August (JJA)

and September-October-November (SON), the a posteriori simulation improves the consistency between model and observation

especially in DJF and SON. The a posteriori model simulated AOD in JJA is still lower than MODIS aerosol products over India. The high aerosol loading is occurred over eastern China throughout the year, which can be inferred both from observations and simulations. While the a posteriori simulated AOD over eastern China is lower than MODIS data in JJA. Over biomass burning regions (e.g. southern Africa and South America), the model simulation shows a consistent seasonal variability with MODIS data

that the peaks are in JJA and SON. One of the major discrepancies between the a priori model simulation and MODIS over biomass burning regions is that the a priori model simulated AOD is lower than observations. The biomass burning peak over South America is observed in SON by MODIS, however the a priori model simulated AOD in JJA is higher than that in SON. The a posteriori simulation using retrieved emissions reduces this bias. The 1-2 month delayed biomass burning peak inferred from observations has also been reported over Africa in recent study by Zheng et al. (2018). Overall, the a posteriori model

simulated AOD shows a better agreement with independent MODIS observations over southern Africa and South America, where the aerosol are associated with biomass burning emissions. Over central Europe, MODIS observed a high aerosol loading event in JJA (wide fires events over central Russia in 2010 summer; see Figure 8), which is not well reproduced by a priori GEOS-Chem simulation. The retrieved emission data help to improve the a posteriori simulation reporting high aerosol loading there, however the a posteriori AOD is still somewhat lower than MODIS.

The statistics of a priori and a posteriori GEOS-Chem simulated AOD evaluated with MODIS AOD are shown in Table 3. The evaluation was conducted at 550 nm for daily products in year 2010. The GEOS-Chem simulation shows a better agreement with independent MODIS AOD from a priori (R=0.49, RMSE=0.19, MAE=0.054, NMB=-18.6%) to a posteriori (R=0.59, RMSE=0.15, MAE=0.045, NMB=-13.2%) simulation.

**Table 4. Statistics for evaluation of a priori and a posteriori daily GEOS-Chem simulation with MODIS AOD and OMI AAOD in year 2010**

|  | A priori GEOS-Chem | A posteriori GEOS-Chem |
|---|---|---|
| MODIS AOD | R=0.49; RMSE=0.19; MAE=0.054; NMB=-18.6% | R=0.59; RMSE=0.15; MAE=0.045; NMB=-13.2% |
| OMI AAOD | R=0.39; RMSE=0.026; MAE=0.015; NMB=-17.8% | R=0.38; RMSE=0.014; MAE=0.012; NMB=19.2% |

**3.3.3 Comparison with OMI**

In this section, we discuss the seasonal variability of AAOD from the Ozone Monitoring Instrument (OMI) near-UV algorithm (OMAERUV) and the a priori and a posteriori GEOS-Chem simulation, as well as their differences. The OMAERUV algorithm uses two UV wavelengths to derive columnar AAOD (Torres et al., 2007). The climatology of Atmospheric Infrared

Sounder (AIRS) carbon monoxide (CO) observations and aerosol layer height information from the Cloud-Aerosol Lidar with Orthogonal Polarization (CALIOP) are adopted in the latest OMAERUV algorithm (Torres et al., 2013), and the assessment of the OMI/OMAERUV aerosol products are described in Ahn et al. (2014) and Jethva et al. (2014). The OMI/OMAERUV Level 3 aerosol products with 1-degree spatial resolution are used for model evaluation. To aggregate to model resolution, any grid box 5 with less than 2 OMI/OMAERUV retrievals is omitted.

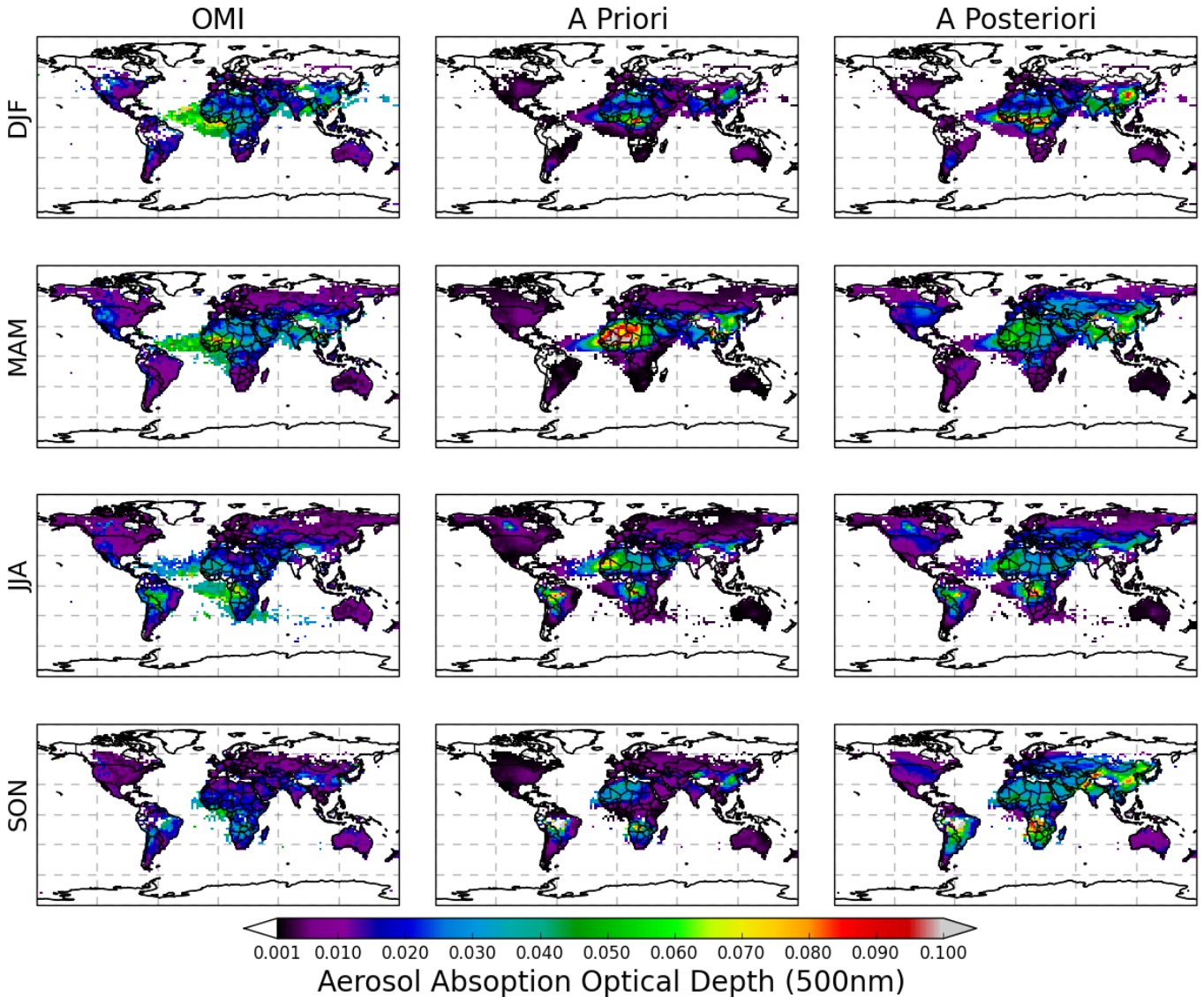

**Figure 11. Comparison of a priori and a posteriori GEOS-Chem simulated seasonal AAOD at 500 nm with OMI/OMAERUV products**

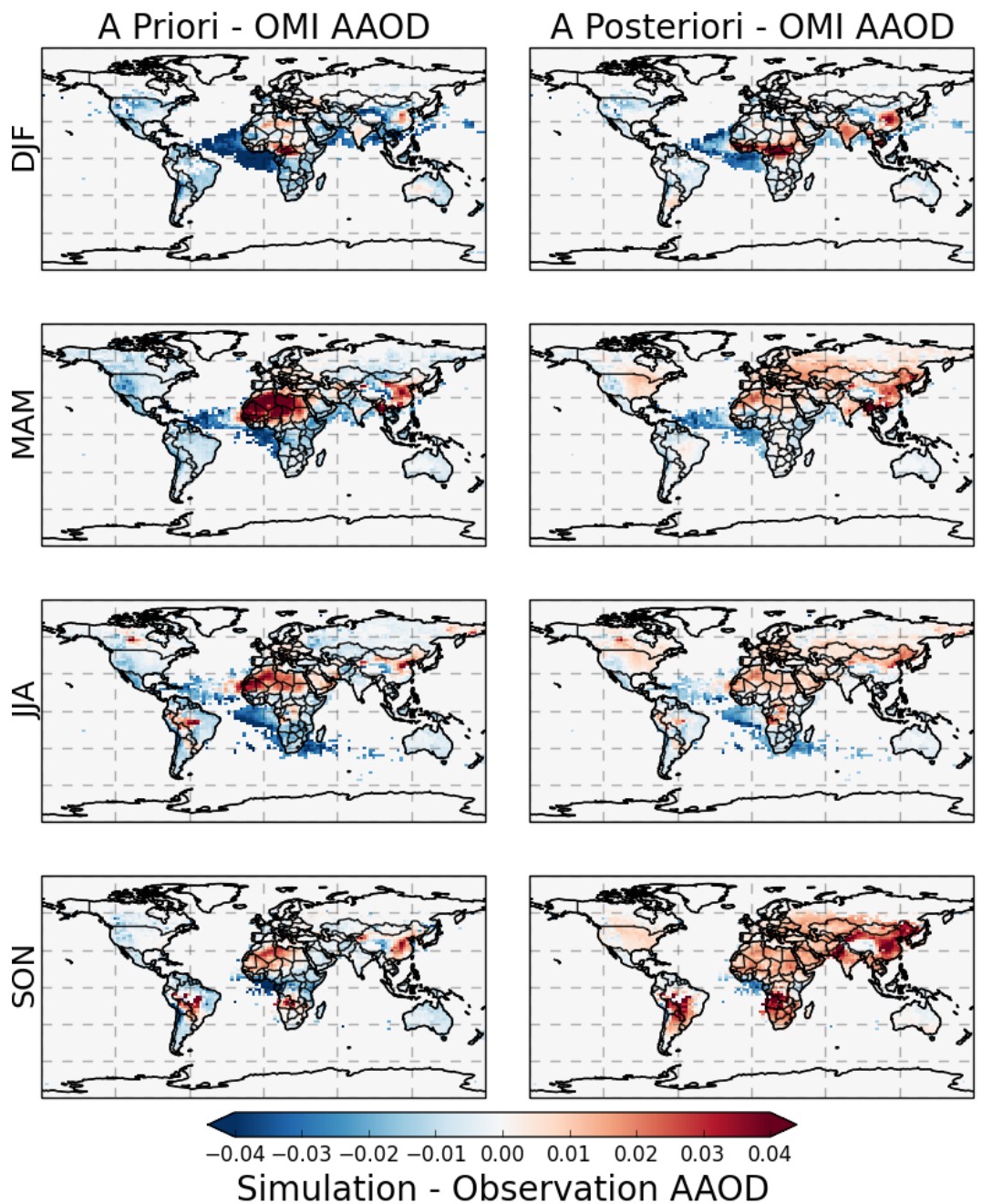

**Figure 12. Differences between a priori and a posteriori GEOS-Chem seasonal AAOD with OMI/OMAERUV products**

Figure 11 shows seasonal variability of AAOD at 500 nm for OMI/OMAERUV (left panels), a priori GEOS-Chem (middle panels), and a posteriori GEOS-Chem (right panels). The differences between a priori and a posteriori GEOS-Chem AAOD with OMI are presented in Figure 12. Here, the GEOS-Chem 500 nm AAOD is interpolated using AAOD at two wavelengths based on Absorbing Ångström exponent, $\alpha_a(443-865) = \ln\left(\frac{\tau_a(443)}{\tau_a(865)}\right)/\ln\left(\frac{865}{443}\right)$. Consistent with the simulations of AOD, the a priori GEOS-Chem simulated AAOD over Sahara is higher than OMI AAOD products throughout the year. However, the a priori AAOD is lower than OMI data in the westward grid boxes over the Atlantic ocean, which suggests that the

removal processes in the model may be too rapid (Ridley et al., 2012, 2016). The a posteriori GEOS-Chem AAOD over the Sahara source region decreases (with respect to a priori AAOD) and shows a good match with OMI AAOD; nevertheless the discrepancy for transported dust over ocean grid boxes still exists.

The a posteriori GEOS-Chem simulation indicates moderate aerosol absorption (AAOD$_{500nm}$ = ~0.05) over central and

northern China in SON, whereas the a priori model simulation and OMI data show less aerosol absorption (AAOD$_{500nm}$ = ~0.02). The a posteriori GEOS-Chem simulation shows good agreement with OMI over South America in JJA, but the a priori AAOD is higher than both a posteriori AAOD and OMI. Seasonal biomass burning events can cause high AAOD in the OMI observations as well as the a priori and a posteriori GEOS-Chem simulations over southern Africa and South America; however the a posteriori AAOD is higher than the a priori and OMI AAODs. Over India, OMI-observed aerosol absorption shows a peak

season in March-April-May (MAM), and the a posteriori simulation successfully reproduces this seasonal peak. However, the a posteriori AAOD is slightly higher than OMI in DJF and SON over India. This is partially due to the fact that the inversion framework was fixed to SU as a priori emission database, which could result in the propagation of the uncertainty from a priori SU into a posteriori BC and OC emission over regions with predominant SU source.

The statistics of a priori and a posteriori GEOS-Chem daily AAOD evaluated with OMI AAOD are shown in Table 3.

The a priori AAOD shows a similar correlation coefficient (R=0.39) to the a posteriori AAOD (R=0.38) and similar mean absolute error (a priori: MAE=0.015; a posteriori: MAE=0.012). However, the a posteriori AAOD presents a better root mean square error (a priori: RMSE=0.026; a posteriori: RMSE=0.014). Meanwhile, the normalized mean bias is significant for both the a priori (NMB=-17.8%) and a posteriori (NMB=19.2%) simulations, which indicates that both GEOS-Chem AAOD simulations differ from OMI by ~20%.

### 3.3.4 Comparison with IMPROVE surface measurements of BC concentration

This section describes an evaluation that we conducted for simulated surface concentrations of BC mass. The Interagency Monitoring of Protected Visual Environments (IMPROVE) is a network of in situ aerosol measurement sites located in US National parks (Malm et al., 1994, 2003). We obtained the surface concentration measurements of BC at IMPROVE sites,

and evaluate the a priori and a posteriori GEOS-Chem simulation of surface BC at these sites for year 2010. Results for the annual mean surface concentration of BC are shown in Figure 13. Here, we see that the a priori GEOS-Chem surface BC concentration is lower than the IMPROVE data, with NMB=-26.20%, MB=-0.051, which is consistent with previous work (Chin et al., 2014; Wang et al., 2014). The a posteriori GEOS-Chem simulation has higher values and therefore shows better agreement with the high surface BC concentrations over the eastern US than the a priori simulation. However, the a posteriori GEOS-Chem

simulation overestimates BC concentrations over most of the central and western US sites where surface BC concentrations are

low; this leads to an ensemble NMB=+18.29%, MB=+0.035. In terms of correlation coefficient, the a posteriori simulation slightly improves it from 0.43 to 0.47.

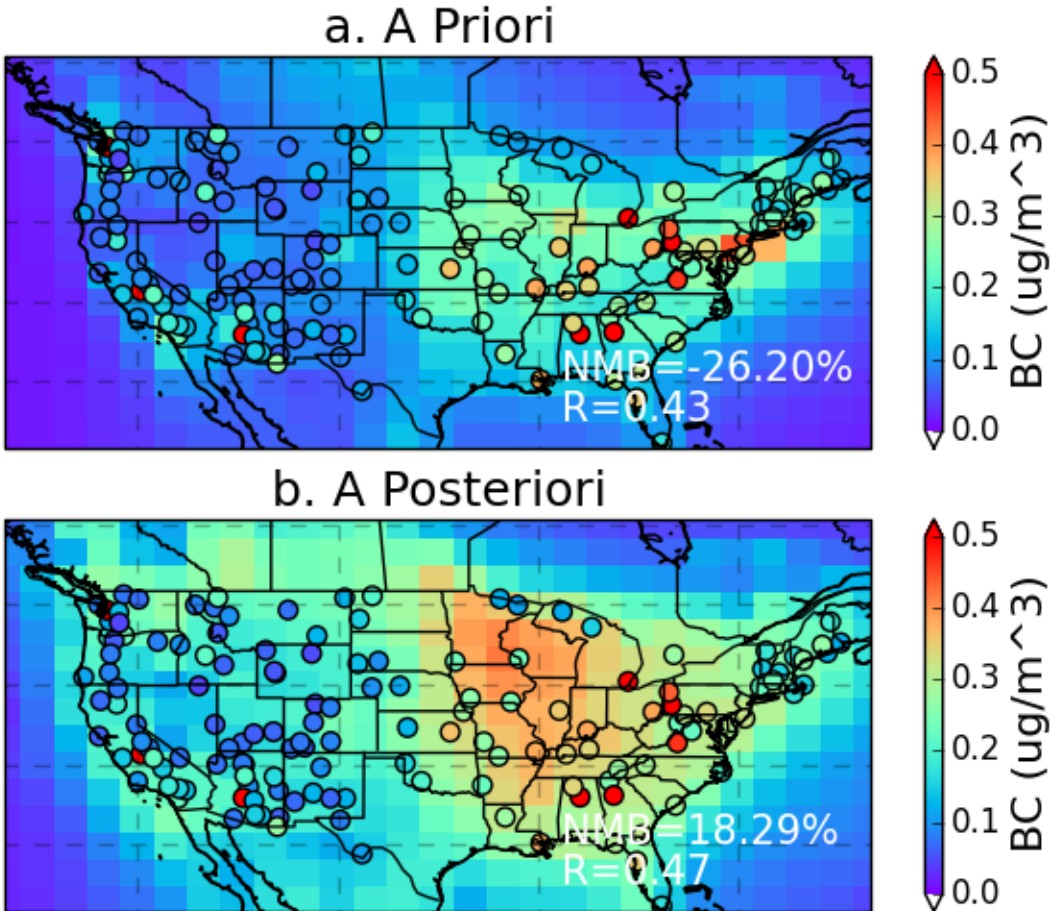

**Figure 13. Evaluation a priori (a) and a posteriori (b) GEOS-Chem simulated annual mean surface BC concentration in the Unite States with observations (circles) from IMPROVE network for year 2010. The correlation coefficient (R) and normalized mean bias (NMB) are provided in the right corner.**

## 4 Discussion and conclusions

In this study, we have used PARASOL spectral AOD and AAOD generated by the GRASP algorithm to retrieve global BC, OC and DD aerosol emissions based upon the development of the GEOS-Chem inverse modelling framework. Specifically, PARASOL/GRASP AOD and AAOD at 6 wavelengths (443, 490, 565, 670, 865 and 1020 nm) were used to correct the aerosol emission fields using the inverse modelling framework developed by Chen et al. (2018). This resulted in improved global daily aerosol emissions at the spatial resolution of transport models.

The retrieved global annual DD emission of 731.6 Tg/yr is 42.4% less than the a priori GEOS-Chem DD model of 1269.4 Tg/yr. The retrieved DD annual emission is also near the low end of the estimates provided by the AeroCom project (700-4000 Tg/yr; Huneeus et al., 2011; Kinne et al., 2006; Textor et al., 2006). This is partially due to the exclusion of super coarse mode dust particles in the retrieval, as only particle radii ranging from 0.1 to 6.0 $\mu m$ are taken into account. An overestimation of simulated global DD emission was also reported by Huneeus et al. (2012), who assimilated MODIS AOD into a global aerosol model. However, the overestimation of dust lifetime, poor characterization of soil data used for dust mobilization calculation, and overestimation of small dust particles (Kok et al., 2017) are also non-negligible.

We used the GFED v4s and HTAP v2 emission inventories for the a priori GEOS-Chem simulation, where the global BC emission is 6.9 Tg/yr. The retrieved global BC emission is 18.4 Tg/yr, which is 166.7 % higher than the a priori model inventories. Our estimation of global BC emission is close to the other top-down studies by Huneeus et al. (2012) and Cohen and Wang (2014). The former study estimates global BC emission 15 Tg/yr, whereas the latter gives the emission range from 14.6 to 22.2 Tg/yr. In comparison, the best estimate from bottom-up inventory methods is 7.5 Tg/yr, with an uncertainty range of 2.0 to 29.0 Tg/yr (Bond et al., 2013). Correspondingly, the retrieved global annual OC emission flux is 109.9 Tg/yr, which is 184.0 % higher than a priori emission inventories (38.7 Tg/yr). Huneeus et al. (2012) estimated global organic matter (OM) emission as 119 Tg/yr, which is equivalent to ~85 Tg/yr OC based upon a conversion factor of 1.4 for OM/OC.

We introduced a method to separate anthropogenic and biomass burning BC and OC from retrieved total BC and OC emissions by using a priori daily proportion. The retrieved anthropogenic BC emission is 14.8 Tg/yr and 4.6 Tg/yr, which is 217.3% higher than a priori inventory adopted from HTAP v2. The retrieved biomass burning BC emission is 4.6 Tg/yr, showing an increase of 56.5% with respect to a priori GFED v4s emission inventory in GEOS-Chem model. The retrieved total OC emission is split into anthropogenic OC 85.6 Tg/yr and biomass burning OC 24.3 Tg/yr, which are 357.8% and 21.5% higher than the a priori emission inventories.

The resulting GEOS-Chem a posteriori annual mean AOD and AAOD using the retrieved emission data are 0.119 and 0.0071 respectively at the 550 nm wavelength. These calculations indicate a decrease of 8% for AOD and an increase of 69% for AAOD with respect to AeroCom Phase II multi model assessment.

The fidelity of the results is confirmed by evaluating the a posteriori simulations of aerosol properties with independent measurements. Namely, in order to validate the retrieved emissions, the a posteriori model simulations of AOD, AAOD, SSA, AExp and AAExp were compared to independent measurements from AERONET, MODIS and OMI. We also note that the AERONET dataset is temporally more frequent than the PARASOL observations that we used to obtain the a posteriori emissions. The a posteriori GEOS-Chem daily AODs and AAODs show a better agreement (higher correlation coefficients and lower biases) with AERONET values than the a priori simulation. In addition, the a posteriori SSA, AExp, and AAExp also

show good agreement with AERONET data; this indicates that PARASOL provided sufficient constraints for fitting spectral AOD and AAOD, and that the retrieved emission dataset can provide reliable model simulations. Besides, a posteriori GEOS-Chem AOD and AAOD exhibit a similar seasonal pattern with the MODIS AOD and OMI AAOD respectively during all seasons, which indicates that the retrieved emissions are capable of capturing the major events (e.g. dust hot spots, biomass burning, anthropogenic activities). However, the a posteriori simulation overestimates AOD and AAOD over that Sahara dust source region, while underestimating AOD and AAOD over grid boxes located downwind over the Atlantic Ocean. This is probably caused by an overestimation of the retrieved DD emission over Sahara combined with a GEOS-Chem removal process that may be too rapid (Ridley et al., 2012, 2016). During biomass burning seasons (e.g. JJA and SON over South America and southern Africa), the a posteriori AOD shows good agreement with MODIS; meanwhile the a posteriori AAOD is slightly higher than the OMI AAOD. This could be caused by light-absorbing OC that is not included in the simulation or the inversion. Light-absorbing OC is also known as BrC, and it is characterized by absorption that decreases from UV to mid-visible wavelengths (Feng et al., 2013; Lack et al., 2012). The lack of BrC in our framework could cause the retrieval to generate more BC in order to capture the observed aerosol absorption (since BC has strong absorption throughout the visible and near-infrared wavelengths; Sato et al., 2003). As a consequence, the a posteriori AAOD may be overestimated from mid-visible to near infrared wavelengths when BrC is not included.

Our evaluation of BC surface concentrations with the IMPROVE network over the United States indicates the possibility to improve the aerosol mass simulation based on inversion of satellite-derived columnar spectral aerosol extinction and absorption. However, the a posteriori simulation shows overestimation of surface BC concentration over sites with low levels of BC, which is probably due to an overestimation of a posteriori BC emissions over low-loading regions, or a modelled BC lifetime that is too long (Lund et al., 2018; Wang et al., 2014).

The result of this study is the satellite-based aerosol emission database that is commonly used as GEOS-Chem default input and was adjusted to tune the global POLDER/PARASOL observations of spectral AOD and AAOD. The analysis in the paper, as well as previous work by Chen et al. (2018) showed that the aerosol distribution modelled with the satellite-based aerosol emission improves the agreement of modelling results with the independent AERONET, MODIS, OMI and IMPROVE data. These validation results support the validity of the identified corrections of the emissions. Therefore, if this database is used for initialization of not only GEOS-Chem model but any other aerosol transport or GCM (Global Climate Modelling), then the effects of these suggested significant corrections to the amount of mass of DD, BC and OC and their time and space on the climate and environment can be studied.

To recapitulate, we derived global BC, OC and DD aerosol emission fields in a GEOS-Chem modelling framework that was constrained with PARASOL/GRASP spectral AODs and AAODs. Our study shows that this method can be useful for

improving global aerosol simulations with CTMs. In the future, we plan to use the entire PARASOL dataset to generate a satellite-based aerosol emission database; this is expected to improve multi-year aerosol simulations of AOD, AAOD, SSA, AExp, and AAExp in CTMs. In addition, the efforts to better understand of aerosol life cycle at process level (Textor et al., 2007) are essential to inversion of aerosol emission, aerosol prediction and aerosol climate effect evaluation.

**Data availability**

The AERONET version 2 data is available at https://aeronet.gsfc.nasa.gov/. The MODIS C6 merged AOD and OMI/OMAERUV AAOD products are publicly available at NASA's Goddard Earth Sciences Data and Information Services Center (GES DISC, https://disc.gsfc.nasa.gov/datasets/). The IMPROVE surface BC concentration data is available at

http://vista.cira.colostate.edu/Improve/. The retrieved daily DD, BC and OC aerosol emissions for year 2010 are publicly available at http://www-loa.univ-lille1.fr/article/a/emissions-aerosols-echelle-globale-restituees-par-modelisation-inverse-Chen-et-al.

**Author contribution**

CC, OD, DKH and TL contributed to the inversion algorithm development. DF, PL, AL prepared the aerosol dataset from POLDER/PARASOL generated by the GRASP algorithm. CC, TL and FD carried out the data processing. CC analysed the results with contribution from OD, MC, DKH and GLS; and LL, QH, PL, AL and BT participated in science discussions. CC and OD wrote the paper with inputs from all authors.

**Competing interests**

The authors declare that they have no conflict of interest.

**Acknowledgements**

This work is supported by the Laboratory of Excellence CaPPA – Chemical and Physical Properties of the Atmosphere – project, which is funded by the French National Research Agency (ANR) under contract ANR-ll-LABX-0005-01. We would like to

thank the AERONET, MODIS, IMPROVE team for sharing the data used in this study.

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
