# Peer review of "Constraining global aerosol emissions using POLDER/PARASOL satellite remote sensing observations"

_Atmospheric Chemistry and Physics, 2019_

## Referee Comment (RC1) · Anonymous Referee #1 · 16 Aug 2019

The authors derive simultaneously global and annual emissions of black carbon (BC), organic carbon (OC) and desert dust (DD) by constraining the GEOS-Chem model with POLDER/PARASOL spectral aerosol optical depth (AOD) and aerosol absorption optical depth (AAOD). Emission fluxes of sulphate (SU) and sea salt (SS) are not estimated and remain constant with the inversion. The inversion system applied in this work is an extension of one previously applied on a regional scale and is based on an adjoint of the GEOS-Chem model. The inversion method is applied to the year 2010 and an extensive indirect validation is conducted against independent AOD and AAOD measurements. The simulated AOD and AAOD with the a priori and a posteriori emissions are compared against equivalent observations from AERONET, MODIS and

[Figure]

OMI. In addition, the authors compare the estimated fluxes with estimates from the literature. The estimated emission for BC, OC and DD are 18. Tg/yr, 109.9 Tg/yr and 731.6 Tg/yr, respectively, representing a change of 166.7%, 184.0% and -42.4% with respect to the a priori emissions. The research presented is innovative, very interesting and the paper is well written. I recommend this paper to be published in ACP after some minor comments have been addressed.

General Comments:

1. Although the authors describe the inversion system in general terms and present references providing further description of the system, they could give more information making it easier for the reader to understand the system. For instance, the authors did not specify whether the 7 DD bins in the model are perturbed homogeneously or not. It seems that they are but this should be made clear to the reader. Also, although the authors define the diagonal terms in both covariance error matrices it is not stated clearly whether both are actually diagonal matrices, one assumes they are since this is often the case, but again this should be stated explicitly. Finally, although the authors conduct an extensive validation of the inversion system and also compare the simulated AOD and AAOD between the a priori and a posteriori simulation, at no point do they actually show the POLDER products that are used to constrain the model and the improvements of the retrieved fields with respect to these products. Although one can only assume that the estimated emissions improve the simulated fields with respect to the inverted observations, presenting maps of the POLDER AOD and AAOD could contribute to understand the differences with the independent observations used in the validation. These maps could be included as supplement material if the authors prefer not to increase the number of figures in the manuscript and the global average could also be added to Table 2. 2. The inversion system applied has the advantage of being able of identifying new sources. This is a "nice feature" of the system since it could provide missing sources not included in the a priori emissions. However, the authors do not indicate whether the system actually identifies any missing source, only that

emissions are reduced or increased. It would be interesting to know whether the initial emissions miss any source present in the final estimate., in particular for the HTAP inventory used as a priori. 3. In section 4 the authors compare the retrieved total BC and OC emissions with estimates found in the literature. However given that multiple biomass burning and anthropogenic emission inventories exist, I would suggest that in addition of comparing the total fluxes as they do, they also compare them separately against biomass burning and anthropogenic inventories.

Specific Comments:

Page 5 line 25: remove "are" from the beginning of the sentence.

Page 8 line 6: I do not consider that seasonal BC and OC variations between a priori and a posteriori emissions can be called similar. Although the second maxima (Aug & Sept) is also observed in the a posteriori, this is not the case for the first maxima where there is a shift between both emission fluxes; while the a priori peaks in March, the a posteriori does so in Apr-May. The authors should better describe the differences and similarities between a priori and a posteriori in this figure.

Page 23 line 9: change "has also reported" to "has also been reported"

Page 23 line 10: review formulation of the sentence after the coma; "where are".

Page 23 line 17: section 3.3 indicates the statistics that will be used in the evaluation with independent measurement and the RMSE used in this section is not included. Why the change? I suggest either the RMSE is included when presenting the statistics in section 3.3 or here the analysis is limited to the statistics presented in that section.

Page 26 line 9: remove "the" between "OMI-observed" and "aerosol".

Page 28 line 28: Replace "probability" with "probably" and move "the" before "retrieved".

Page 29 line 3: A parenthesis is missing, most likely before Feng et al.

---

## Referee Comment (RC2) · Anonymous Referee #2 · 19 Aug 2019

This manuscript presents an interesting approach to improving estimates of global aerosol emissions for use in global modelling, using the model adjoint to facilitate the minimization of a cost function against satellite-retrieved aerosol optical depth (AOD) and absorption aerosol optical depth (AAOD). Starting from a standard emission inventory as prior, this allows an *a posteriori* emission data set to be derived which improves the match of the resulting model to the chosen observations. The manuscript is sound and well presented, and merits publication in ACP, provided the comments below can be addressed.

[Figure]

**Main comments**

1. The whole procedure here is based on a single global model which is used for optimizing the emissions, and then the same model is used to evaluate the resulting dataset. Implicit in this is that the *a posteriori* emissions are tailored to this specific model: some of the changes from the *a priori* emissions may represent corrections of genuine errors in the emissions, while other changes may instead be compensating for model errors (for example, increased/decreased emissions to balance over/under-estimation of removal rates). This is acknowledged in passing (e.g. p.28, lines 27–29), but its implications for the applicability of the *a posteriori* data set should be discussed further. In particular, it should be made explicit in the manuscript whether this is presented as being an improved emission data set for use in this particular global model, or for more general use (in which case some justification for its wider applicability is required).

2. In relation to both the PARASOL/GRASP data used for the optimization, and the MODIS and OMI data used for evaluation, the possible impact of spatial sampling biases is briefly mentioned, but no attempt is made to quantify this. There is also no mention of the temporal resolution of the data, nor the additional impact of temporal sampling biases due to fixed satellite overpass times. (Are temporal means from the model used, or are model values temporally collocated to the satellite overpass against which they are compared? Might unaggregated Level 2 products allow for better collocation with the model?)

**Additional minor comments**

**p.2, lines 3–4.** This suggests that "harmonizing" emissions between different models is a good thing; however this is only true if there is confidence that they are

converging on some kind of "truth". Merely adopting similar emissions without reducing their possible errors is likely to result in the multi-model ensemble of AOD and AAOD becoming under-dispersive.

**p.5, lines 5–7.** This seems to be assuming that it is the values for *uncoated* BC which are applicable, despite the fact that much of the BC in the environment is coated with sulfate, organics or other species. Some justification for the reliance on uncoated properties should be given.

**p.5, line 25.** Spurious "are" in "We used anthropogenic emissions are from the. . . "

**p.5, line 25.** A citation should be given for the "HTAP2 emissions" if possible.

**p.6, line 5.** Please clarify whether 0.05 here is an absolute or relative error. (Because AOD is dimensionless, it can't be obviously inferred from the units.) It's worth making it explicitly clear if the error is uniform, or dependent on the AOD (as is the case for some common retrievals).

**p.7, Table 1.** If $SO_2$ and SS are included in the table, it should be recapitulated in the caption that these are not subject to optimization/refinement in the work presented here, and that this is why the values are necessarily unchanged in the *a posteriori* data set.

**p.10, line 23.** The notation "BC-0.03; OC-0.11" is confusing, as the hyphen is easily misinterpreted as a minus sign attached to the number.

**p.16, Figure 6.** The scatter plots are very unclear, since there are a very large number of overlapping data points in the bulk of the data. Perhaps a density plot would be more appropriate. Also, AOD has a strongly skewed distribution (much closer to lognormal than normal) and the distribution might be clearer if presented on logarithmic axes.

**p.17, lines 5–6.** It is briefly mentioned here that bias increases in some cases (this is true of NMB for AOD, SSA and AAExp in the left column of Figure 6; and MB for AOD and SSA in the right column). However, the authors do not really discuss the reasons *why* the process of optimizing the emissions is leading to a worsening of the bias. These reasons should be explored further in the discussion.

**p.19, lines 8–9** While 202/282 sites improved for AOD is a strong result, the other figures are lower and by AAExp (84/167) it is only half the sites which are improved which is pretty much a null result (unless the improvements on these sites are more substantial than the degradation at others). These figures should be reframed to make clear which of these are significant results, preferably with reference to a clear statement of statistical significance.

**p.25, Figure 12.** There are significant deteriorations (leading to strong positive bias) in the *a posteriori*, particularely over Asia in MAM and SON, and in Eastern/Southern Africa and South America in SON. These need to be more clearly referred to in the text, with some discussion of the likely causes.

---

## Author Comment (AC1) · 5 Oct 2019

We would like to thank the two referees for their time reviewing the manuscript, and for the helpful feedback provided. Please see the attached supplement for our responses to both referees.

**Reviewer #1:**

*The authors derive simultaneously global and annual emissions of black carbon (BC), organic carbon (OC) and desert dust (DD) by constraining the GEOS-Chem model with POLDER/PARASOL spectral aerosol optical depth (AOD) and aerosol absorption optical depth (AAOD). Emission fluxes of sulphate (SU) and sea salt (SS) are not estimated and remain constant with the inversion. The inversion system applied in this work is an extension of one previously applied on a regional scale and is based on an adjoint of the GEOS-Chem model. The inversion method is applied to the year 2010 and an extensive indirect validation is conducted against independent AOD and AAOD measurements. The simulated AOD and AAOD with the a priori and a posteriori emissions are compared against equivalent observations from AERONET, MODIS and OMI. In addition, the authors compare the estimated fluxes with estimates from the literature. The estimated emission for BC, OC and DD are 18. Tg/yr, 109.9 Tg/yr and 731.6 Tg/yr, respectively, representing a change of 166.7%, 184.0% and -42.4% with respect to the a priori emissions. The research presented is innovative, very interesting and the paper is well written. I recommend this paper to be published in ACP after some minor comments have been addressed.*

**Response:**

We thank the referee for the positive and insightful comments. Our point-by-point responses to reviewer's general and specific comments are presented below. The changes to the initial manuscript text and supplement illustrations are marked in red.

*General Comments:*

*1. Although the authors describe the inversion system in general terms and present references providing further description of the system, they could give more information making it easier for the reader to understand the system. For instance, the authors did not specify whether the 7 DD bins in the model are perturbed homogeneously or not. It seems that they are but this should be made clear to the reader. Also, although the authors define the diagonal terms in both covariance error matrices it is not stated clearly whether both are actually diagonal matrices, one assumes they are since this is often the case, but again this should be stated explicitly. Finally, although the authors conduct an extensive validation of the inversion system and also compare the simulated AOD and AAOD between the a priori and a posteriori simulation, at no point do they actually show the POLDER products that are used to constrain the model and the improvements of the retrieved fields with respect to these products. Although one can*

*only assume that the estimated emissions improve the simulated fields with respect to the inverted observations, presenting maps of the POLDER AOD and AAOD could contribute to understand the differences with the independent observations used in the validation. These maps could be included as supplement material if the authors prefer not to increase the number of figures in the manuscript and the global average could also be added to Table 2.*

**Response:**

Thanks for your suggestion. We have added one paragraph in section 2.2 to clarify them. We also included the method to estimate anthropogenic contribution from our retrieved BC and OC emission database, which is also related to the 2nd and 3rd comments.

"The inversion system derives daily total BC, OC and DD aerosol emissions for each grid box. The daily ratio between biomass burning and anthropogenic contribution for BC and OC, and the proportion of DD 7 bins for each grid box is kept as a priori GEOS-Chem assumption. Distinguishing anthropogenic contribution from total emission is crucial for climate effects evaluation. Here, we propose a simple method to estimate daily anthropogenic BC ($E_{BC\_INV\_AN}$) and OC ($E_{OC\_INV\_AN}$) emission from our retrieved total emission ($E_{BC\_INV}$) by using daily proportion of anthropogenic emission over each grid box from a priori emission database:

$$E_{BC\_INV\_AN}(x,y,t) = \frac{E_{BC\_AN}(x,y,t)}{E_{BC\_AN}(x,y,t) + E_{BC\_BB}(x,y,t)} * E_{BC\_INV}(x,y,t) \qquad (2)$$

$$E_{OC\_INV\_AN}(x,y,t) = \frac{E_{OC\_AN}(x,y,t)}{E_{OC\_AN}(x,y,t) + E_{OC\_BB}(x,y,t)} * E_{OC\_INV}(x,y,t) \qquad (3)$$

where $E_{BC\_INV}(x,y,t)$ and $E_{OC\_INV}(x,y,t)$ represent retrieved total BC and OC emission. $E_{BC\_INV\_AN}(x,y,t)$ and $E_{OC\_INV\_AN}(x,y,t)$ are derived anthropogenic BC and OC emissions from retrieved total BC and OC emission database. $E_{BC\_AN}(x,y,t)$ and $E_{OC\_AN}(x,y,t)$ represent anthropogenic BC and OC emission from HTAP v2 database, $E_{BC\_BB}(x,y,t)$ and $E_{OC\_BB}(x,y,t)$ are BC and OC emitted from biomass burning adapted from GFED v4s database. $x, y$ and $t$ indicate index of longitude, latitude and time."

We have added Figure S1 into the supplement materials. Figure S1 shows the global mean POLDER 2°x2.5° AOD and AAOD at 443, 490, 565, 670, 865 and 1020 nm used in the emission inversion. In Table 2, we want to report the global mean component-level

AOD and AAOD from model simulation based on a priori and a posteriori emission datasets. While, satellite aerosol products may have different pixel samplings over a year due to different orbits and swaths. Hence, we hope to keep the comparison between a priori and a posteriori simulation.

[Figure]

Figure S1. Global distribution of PARASOL/GRASP 2°x2.5° spectral AOD and AAOD

at 443, 490, 565, 670, 865 and 1020 nm in year 2010

*2. The inversion system applied has the advantage of being able of identifying new sources. This is a "nice feature" of the system since it could provide missing sources not included in the a priori emissions. However, the authors do not indicate whether the system actually identifies any missing source, only that emissions are reduced or increased. It would be interesting to know whether the initial emissions miss any source present in the final estimate., in particular for the HTAP inventory used as a priori.*

**Response:**

Thanks for reviewer's suggestion. We have added a description of the method used to separate anthropogenic BC and OC emissions from retrieved total BC and OC in section 2.2. The spatial distribution anthropogenic BC and OC emissions and their differences with a priori GEOS-Chem database from HTAP v2 are shown in Supplement illustrations. Globally, the retrieved anthropogenic emissions are 14.8 Tg/yr for BC and 85.6 Tg/yr for OC, representing an increasing of 217.3% for BC and 357.8% for OC with respect to the a priori emission database. We have added discussions of retrieved anthropogenic and biomass burning BC and OC emissions in section 3.1 and one table to summarize the global values.

"Table 2 compares the retrieved annual anthropogenic and biomass burning BC and OC emissions with a priori emission database in GEOS-Chem. The method used to separate anthropogenic from total emission is described in Section 2.2. The retrieved anthropogenic emissions are 14.8 Tg/yr for BC and 85.6 Tg/yr for OC, representing an increasing of 217.3% for BC and 357.8% for OC. Meanwhile, the retrieved biomass burning emissions of BC and OC are 3.6 Tg/yr and 24.3 Tg/yr, corresponding to an increase of 56.5% and 21.5% with respect to the a priori emission database. The comparison of spatial distribution of anthropogenic and biomass burning emission of BC and OC are presented in the supplement illustrations in the Figures S5 and S6."

Table 2. Anthropogenic (AN) and biomass burning (BB) emissions (unit: Tg/yr) of BC and OC in year 2010

|  | BC | | OC | |
|---|---|---|---|---|
|  | AN | BB | AN | BB |

| | | | | |
|---|---|---|---|---|
| *A Priori* | 4.6 | 2.3 | 18.7 | 20.0 |
| *A Posteriori* | 14.8 | 3.6 | 85.6 | 24.3 |

[Figure]

Figure S5. Global distribution of BC emissions in 2010 for total BC (left panel), anthropogenic BC (middle panel) and biomass burning BC (right panel) based on (a) a priori and (b) a posteriori emission datasets; and the differences between a posteriori and a priori emission datasets (c)

[Figure]

Figure S6. Same as Figure S5, but for OC

In the future, the further analysis in daily and grid box level is needed.

*3. In section 4 the authors compare the retrieved total BC and OC emissions with estimates found in the literature. However given that multiple biomass burning and anthropogenic emission inventories exist, I would suggest that in addition of comparing the total fluxes as they do, they also compare them separately against biomass burning and anthropogenic inventories.*

**Response:**

Thanks for review's constructive suggestion, which can help us improve the value of our study. We have made several changes to include additional retrieved BC and OC emissions from anthropogenic and biomass burning sources (see in 1st and 2nd comments).

*Specific Comments:*

*1. Page 5 line 25: remove "are" from the beginning of the sentence.*

**Response:**

Corrected.

*2. Page 8 line 6: I do not consider that seasonal BC and OC variations between a priori and a posteriori emissions can be called similar. Although the second maxima (Aug & Sept) is also observed in the a posteriori, this is not the case for the first maxima where there is a shift between both emission fluxes; while the a priori peaks in March, the a posteriori does so in Apr-May. The authors should better describe the differences and similarities between a priori and a posteriori in this figure.*

**Response:**

We have rephrased the sentence as "Both the a priori and a posteriori BC and OC emission inventories show a maximum in August and September, while the second peak in March observed in the a priori database shift to April and May in the a posteriori database."

*3. Page 23 line 9: change "has also reported" to "has also been reported"*

**Response:**

Corrected.

*4. Page 23 line 10: review formulation of the sentence after the coma; "where are".*

**Response:**

We have revised this sentence as "Overall, the a posteriori model simulated AOD shows a better agreement with independent MODIS observations over southern Africa and South America, where the aerosol are associated with biomass burning emissions."

*5. Page 23 line 17: section 3.3 indicates the statistics that will be used in the evaluation with independent measurement and the RMSE used in this section is not included. Why the change? I suggest either the RMSE is included when presenting the statistics in section 3.3 or here the analysis is limited to the statistics presented in that section.*

**Response:**

Thanks for the suggestion. We have added the statistics of RMSE in section 3.3.

*6. Page 26 line 9: remove "the" between "OMI-observed" and "aerosol".*

**Response:**

Corrected.

*7. Page 28 line 28: Replace "probability" with "probably" and move "the" before "retrieved".*

**Response:**

Corrected.

*8. Page 29 line 3: A parenthesis is missing, most likely before Feng et al.*

**Response:**

Corrected.

**Reviewer #2:**

*This manuscript presents an interesting approach to improving estimates of global aerosol emissions for use in global modelling, using the model adjoint to facilitate the minimization of a cost function against satellite-retrieved aerosol optical depth (AOD) and absorption aerosol optical depth (AAOD). Starting from a standard emission inventory as prior, this allows an a posteriori emission data set to be derived which improves the match of the resulting model to the chosen observations. The manuscript is sound and well presented, and merits publication in ACP, provided the comments below can be addressed.*

**Response:**

We appreciate the careful review and constructive suggestions. Our point-by-point responses to reviewer's general and specific comments are presented below. The changes to the initial manuscript text and supplement illustrations are marked in red.

*Main Comments:*

*1. The whole procedure here is based on a single global model which is used for optimizing the emissions, and then the same model is used to evaluate the resulting dataset. Implicit in this is that the a posteriori emissions are tailored to this specific model: some of the changes from the a priori emissions may represent corrections of genuine errors in the emissions, while other changes may instead be compensating for model errors (for example, increased/decreased emissions to balance over/under-estimation of removal rates). This is acknowledged in passing (e.g. p.28, lines 27–29), but its implications for the applicability of the a posteriori data set should be discussed further. In particular, it should be made explicit in the manuscript whether this is presented as being an improved emission data set for use in this particular global model, or for more general use (in which case some justification for its wider applicability is required)*

**Response:**

I agree that we cannot ascribe everything to emissions. Previous study by Textor et al. (2007) has shown that the unified emissions can not lead to harmonized model performance for the reason of model diversity of aerosol life cycle at process level. In our previous study over Africa (Chen et al., 2018), we implemented retrieved emission derived from GEOS-Chem model into GEOS-5/GOCART model. The a posteriori model simulation using retrieved emission showed improvement for agreement with independent measurements. However, we cannot ignore that GEOS-Chem and GEOS-5/GOCART are using similar meteorology. In this study, we evaluated a posteriori model simulation with independent measurements and more parameters than fitted from

PARASOL/GRASP, for example, surface concentration from IMPROVE network. A posteriori GEOS-Chem model has performed better than a priori simulation. In the future, we plan to do similar test over global with GEOS-5/GOCART and other global models to further evaluate our emission dataset. We have included discussion of study by Textor et al. (2007) in the conclusions.

*2. In relation to both the PARASOL/GRASP data used for the optimization, and the MODIS and OMI data used for evaluation, the possible impact of spatial sampling biases is briefly mentioned, but no attempt is made to quantify this. There is also no mention of the temporal resolution of the data, nor the additional impact of temporal sampling biases due to fixed satellite overpass times. (Are temporal means from the model used, or are model values temporally collocated to the satellite overpass against which they are compared? Might unaggregated Level 2 products allow for better collocation with the model?)*

**Response:**

Thanks for your suggestion. Yes, I agree that the use of model data according to satellite overpass and un-aggregated satellite Level 2 products could have better agreement between model and satellite observation. In this study, we use MODIS and OMI Level 3 daily products and collect model daily averaged data if the pixel has available satellite data for validation. In order to collocated with model simulation, the Level 3 MODIS and OMI products are aggregated into 2°x2.5° grid box. Any grid box with less than ~50% coverage is omitted. The information has been added in section 3.3.2. Table 4 shows the statistics for evaluation of a priori and a posteriori daily GEOS-Chem simulation with MODIS and OMI. The matched-up criterion is constant for a priori and a posteriori validation.

*Additional minor Comments:*

*1. p.2, lines 3–4. This suggests that "harmonizing" emissions between different models is a good thing; however this is only true if there is confidence that they are converging on some kind of "truth". Merely adopting similar emissions without reducing their possible errors is likely to result in the multi-model ensemble of AOD and AAOD becoming under-dispersive.*

**Response:**

I agree. Previous study by Textor et al. (2007) has shown that the unified emissions can not lead to harmonized model performance for the reason of model diversity of aerosol life cycle at process level. We are not fully optimistic that our retrieved emission database can improve performance of many models, although we are motivated to check this and explore this aspect in the future.

*2. p.5, lines 5–7. This seems to be assuming that it is the values for uncoated BC which are applicable, despite the fact that much of the BC in the environment is coated with sulfate, organics or other species. Some justification for the reliance on uncoated properties should be given.*

**Response:**

Thanks for your suggestion. We have added discussions of the use of uncoated BC in section 2.2. "The assumption of external mixing of spherical particles is adopted in our inversion, as it is commonly done in most of CTMs. It should be noted, however, that the particle morphologies and mixing state could have strong affects on scattering and absorption properties, thus affecting mass to optical conversion (Liu and Mishchenko, 2018). For example, the "lensing effect" of less absorbing components coated on BC could amplify total aerosol absorption (Lesins et al., 2002). The absorption enhancement due to coating is estimated ~1.5 (Bond and Bergstrom, 2006). Recent study by Curci et al. (2019) implemented partial internal mixing for regional simulation, and found it could improve simulation of total absorption while the spectral dependence can not be well reproduced. Therefore, in this approach, as well as, generally in CTMs there is some intrinsic ambiguity in assumptions influencing efficiency of scattering and absorption of aerosol particles. This ambiguity is certainly among of major factors affecting accuracy of derived emissions in the current approach."

*3. p.5, line 25. Spurious "are" in "We used anthropogenic emissions are from the. . . "*

**Response:**
Corrected.

*4. p.5, line 25. A citation should be given for the "HTAP2 emissions" if possible.*

**Response:**

Thanks for your suggestion. We have added two citations for HTAP Phase 2 emission database.

*5. p.6, line 5. Please clarify whether 0.05 here is an absolute or relative error. (Because AOD is dimensionless, it can't be obviously inferred from the units.) It's worth making it explicitly clear if the error is uniform, or dependent on the AOD (as is the case for some common retrievals).*

**Response:**

Thanks for the suggestion. We have rephrased this sentence as "The current understanding of the accuracy of PARASOL/GRASP products is ~0.05 for absolute AOD."

*6. p.7, Table 1. If SO2 and SS are included in the table, it should be recapitulated in the caption that these are not subject to optimization/refinement in the work presented here, and that this is why the values are necessarily unchanged in the a posteriori data set.*

**Response:**

Thanks for your suggestion. We have added the description of $SO_2$ and SS emission in the caption.

*7. p.10, line 23. The notation "BC-0.03; OC-0.11" is confusing, as the hyphen is easily misinterpreted as a minus sign attached to the number.*

**Response:**

Thanks for your suggestion. This sentence has been revised as "On the other hand, the a posteriori simulation indicates significant increase of carbonaceous AOD from a priori 0.014 (BC: 0.003; OC: 0.011) to a posteriori 0.040 (BC: 0.008; OC: 0.032)."

*8. p.16, Figure 6. The scatter plots are very unclear, since there are a very large number of overlapping data points in the bulk of the data. Perhaps a density plot would be more appropriate. Also, AOD has a strongly skewed distribution (much closer to lognormal than normal) and the distribution might be clearer if presented on logarithmic axes.*

**Response:**

Thanks for review's suggestion. We have updated Figure 6 by separating a priori and a

posteriori evaluation with AERONET into two columns and using density plot to avoid data overlapping.

*9. p.17, lines 5–6. It is briefly mentioned here that bias increases in some cases (this is true of NMB for AOD, SSA and AAExp in the left column of Figure 6; and MB for AOD and SSA in the right column). However, the authors do not really discuss the reasons why the process of optimizing the emissions is leading to a worsening of the bias. These reasons should be explored further in the discussion.*

**Response:**

Yes. Although the a posteriori simulation improved AOD correlation coefficient 0.59 to 0.66, we cannot ignore the NMB for AOD changes from a priori 8.15% to a posteriori 21.45, and MB for AOD increase from a priori 0.02 to a posteriori 0.04. The possible reason is that the re-gridded 2°x2.5° PARASOL AOD could have a positive bias in compassion with AERONET point measurements inside ~200 km x ~200 km. In addition, AERONET daily average is observed at daytime, while model daily results are based on daytime and nighttime simulation. We should conduct detailed study of this point in the future. The worsening bias of SSA and AExp is mostly due to that we are not fitting these values directly. We have added some discussion of it in the revised version.

*10. p.19, lines 8–9 While 202/282 sites improved for AOD is a strong result, the other figures are lower and by AAExp (84/167) it is only half the sites which are improved which is pretty much a null result (unless the improvements on these sites are more substantial than the degradation at others). These figures should be reframed to make clear which of these are significant results, preferably with reference to a clear statement of statistical significance.*

**Response:**

Thanks for the suggestion. Figure 7 shows the differences between a posteriori and a priori GEOS-Chem simulated aerosol properties correlation coefficients (R) with AERONET daily aerosol products. If the a posteriori simulation has a better R, the color will be red. Otherwise, it will be blue. This analysis is done over all sites with collocated data in 2010, no matter how many collated points there are. Hence, the AAExp all-points validation in Figure 6 shows improvement of R from a priori 0.01 to a posteriori 0.62; but we only see improvements from 84 sites over total 167 in Figure 7. To improve the

presentation, we have revised Figure 7 by using the size of cycles to represent number of points over each site.

[Figure]

**Figure 7. The differences between a posteriori and a priori GEOS-Chem simulated AOD, AAOD, SSA, AExp and AAExp correlation coefficients (R) with AERONET daily aerosol products over all sites with collocated data in 2010**

*11. p.25, Figure 12. There are significant deteriorations (leading to strong positive bias) in the a posteriori, particularely over Asia in MAM and SON, and in Eastern/Southern Africa and South America in SON. These need to be more clearly referred to in the text, with some discussion of the likely causes.*

**Response:**

Yes, I agree. Indeed, the a posteriori simulated AAOD show strong positive bias over biomass burning and industrial regions at particular seasons in comparison with OMI AAOD. The inversion framework keeps SU as a priori, which could lead to overestimate of BC and OC over industrial area, once the a priori SU is underestimated. Over biomass burning region (Southern Africa and South America in SON), the posteriori simulation of AOD show good agreement with MODIS AOD (Figure 9 and 10). It could be from uncertainties of OMI AAOD products. OMI derives aerosol absorption relying on UV channels, which are more sensitive to coarse mode dust absorption than biomass burning fine mode particles. This explains why the improvements are mainly over dust regions in comparison with OMI. We have added some discussions in section 3.3.3.

**References**

Bond, T. C. and Bergstrom, R. W.: Light Absorption by Carbonaceous Particles: An Investigative Review, Aerosol Sci. Technol., 40(1), 27–67, doi:10.1080/02786820500421521, 2006.

Chen, C., Dubovik, O., Henze, D. K., Lapyonak, T., Chin, M., Ducos, F., Litvinov, P., Huang, X. and Li, L.: Retrieval of desert dust and carbonaceous aerosol emissions over Africa from POLDER/PARASOL products generated by the GRASP algorithm, Atmos. Chem. Phys., 18(16), 12551–12580, doi:10.5194/acp-18-12551-2018, 2018.

Curci, G., Alyuz, U., Barò, R., Bianconi, R., Bieser, J., Christensen, J. H., Colette, A., Farrow, A., Francis, X., Jiménez-Guerrero, P., Im, U., Liu, P., Manders, A., Palacios-Peña, L., Prank, M., Pozzoli, L., Sokhi, R., Solazzo, E., Tuccella, P., Unal, A., Vivanco, M. G., Hogrefe, C. and Galmarini, S.: Modelling black carbon absorption of solar radiation: combining external and internal mixing assumptions, Atmos. Chem. Phys., 19(1), 181–204, doi:10.5194/acp-19-181-2019, 2019.

Lesins, G., Chylek, P. and Lohmann, U.: A study of internal and external mixing scenarios and its effect on aerosol optical properties and direct radiative forcing, J. Geophys. Res. Atmos., 107(D10), AAC 5-1-AAC 5-12, doi:10.1029/2001JD000973, 2002.

Liu, L. and Mishchenko, M.: Scattering and Radiative Properties of Morphologically Complex Carbonaceous Aerosols: A Systematic Modeling Study, Remote Sens., 10(10), 1634, doi:10.3390/rs10101634, 2018.

Textor, C., Schulz, M., Guibert, S., Kinne, S., Balkanski, Y., Bauer, S., Berntsen, T., Berglen, T., Boucher, O., Chin, M., Dentener, F., Diehl, T., Feichter, J., Fillmore, D., Ginoux, P., Gong, S., Grini, A., Hendricks, J., Horowitz, L., Huang, P., Isaksen, I. S. A., Iversen, T., Kloster, S., Koch, D., Kirkevåg, A., Kristjansson, J. E., Krol, M., Lauer, A., Lamarque, J. F., Liu, X., Montanaro, V., Myhre, G., Penner, J. E., Pitari, G., Reddy, M. S., Seland, Ø., Stier, P., Takemura, T. and Tie, X.: The effect of harmonized emissions on aerosol properties in global models-an AeroCom experiment., Atmos. Chem. Phys., 7(17), 4489–4501, doi: 10.5194/acp-7-4489-2007, 2007.

---

## Author Response (AR2)

We appreciate so much for the editor's valuable and constructive feedback! The point-by-point responses are listed below.

*Reviewer #1 in comment 1 asked that you specify whether the matrices are diagonal. Please respond by adding an explicit statement on this to the manuscript.*

**Response:**

Yes, we assume the covariance error matrices are diagonal. We have added the description in section 2.3.

*In comment 2, reviewer #1 asked about the possibility to identify missing sources. Again, I don't see how you explicitly responded to this.*

**Response:**

Thanks for your suggestion.

Based our analysis, the major findings are underestimation/overestimation of emissions globally and over specific regions. For instance, as shown in the Table 1. Our analysis suggests very significant correction of global aerosol loading: strong overestimation of desert dust emission and underestimation of the emission of OC and BC. This finding mostly relates to the strength of the sources rather than identifying new aerosol sources location. At the same time, if the detailed analysis of the changes in the emission strength is conducted at local scale it is possible that some qualitatively new emissions patterns will be identified that can be considered and interpreted as an identification of missing sources. However, such detailed analysis is rather effort consuming and considered as part of future activities in continuation of current work.

For example, Figure R1 shows one example of local situation analysis. It illustrates a comparison between daily a priori (upper panel) carbonaceous (BC+OC) aerosol emissions with a posteriori (lower panel) emission database during a wildfire event over region (40°N-60°N, 30°E-50°E). The main differences are the emission strength among grid-boxes. The evaluation of simulated AOD and AAOD over AERONET site Moscow_MSU_MO (labeled in red star in Figure R1) is presented in the manuscript Figure 8. Although, both a priori and a posteriori simulations are capable to capture the daily variations, the simulation using a posterior emission database shows better capability to capture the magnitude of aerosol extinction and absorption.

*With respect to reviewer #2's comment 1, I'd like to see a more direct statement on how the improved emission data set could be applied. The reviewer provides ideas on how to describe this.*

**Response:**

Thanks for your suggestion. We have added discussions about the applicability of the improved emission dataset in the section 4.

"The result of this study is the satellite-based aerosol emission database that is commonly used as GEOS-Chem default input and was adjusted to tune the global POLDER/PARASOL observations of spectral AOD and AAOD. The analysis in the paper, as well as previous work by Chen et al. (2018) showed that the aerosol distribution modelled with the satellite-based aerosol emission improves the agreement of modelling results with the independent AERONET, MODIS, OMI and IMPROVE data. These validation results support the validity of the identified corrections of the emissions. Therefore, if this database is used for initialization of not only GEOS-Chem model but any other aerosol transport or GCM (Global Climate Modelling), then the effects of these suggested significant corrections to the amount of mass of DD, BC and OC and their time and space on the climate and environment can be studied. "

[Figure]

Figure R1. The comparison of daily a priori (upper panel) carbonaceous (BC+OC) aerosol emissions with a posteriori (lower panel) emission database during a wildfire event (August 1st to August 7th in 2010) over region (40°N-60°N, 30°E-50°E); the geo-locations of AERONET Moscow_MSU_MO site are apparent as a red star.

[revised manuscript text omitted]